# Functional organization of cytoplasmic inclusion bodies in cells infected by respiratory syncytial virus

Vincent Rincheval[1], Mickael Lelek[2], Elyanne Gault[1,3], Camille Bouillier[1], Delphine Sitterlin[1], Sabine Blouquit-Laye[1], Marie Galloux[4], Christophe Zimmer[2], Jean-François Eleouet[4] & Marie-Anne Rameix-Welti [1,3]

Infection of cells by respiratory syncytial virus induces the formation of cytoplasmic inclusion bodies (IBs) where all the components of the viral RNA polymerase complex are concentrated. However, the exact organization and function of these IBs remain unclear. In this study, we use conventional and super-resolution imaging to dissect the internal structure of IBs. We observe that newly synthetized viral mRNA and the viral transcription anti-terminator M2-1 concentrate in IB sub-compartments, which we term "IB-associated granules" (IBAGs). In contrast, viral genomic RNA, the nucleoprotein, the L polymerase and its cofactor P are excluded from IBAGs. Live imaging reveals that IBAGs are highly dynamic structures. Our data show that IBs are the main site of viral RNA synthesis. They further suggest that shortly after synthesis in IBs, viral mRNAs and M2-1 transiently concentrate in IBAGs before reaching the cytosol and suggest a novel post-transcriptional function for M2-1.

[1] UMR1173, INSERM, Université de Versailles St. Quentin, Montigny le Bretonneux 78180, France. [2] Institut Pasteur Unité Imagerie et Modélisation, CNRS UMR 3691; C3BI, USR 3756, IP CNRS, Paris 75015, France. [3] AP-HP, Laboratoire de Microbiologie, Hôpital Ambroise Paré, Boulogne-Billancourt 92104, France. [4] Unité de Virologie et Immunologie Moléculaires (UR892), INRA, Université Paris-Saclay, Jouy-en-Josas 78352, France. Correspondence and requests for materials should be addressed to M.-A.R.-W. (email: marie-anne.rameix-welti@uvsq.fr)

Human respiratory syncytial virus (RSV) is the primary cause of severe respiratory infection in infants worldwide[1]. RSV also represents a significant burden for the elderly, asthmatic and immunocompromised patients, and other at-risk populations[2]. RSV is an enveloped, non-segmented, negative-sense RNA virus that belongs to the *Orthopneumovirus* genus in the *Pneumoviridae* family of the order *Mononegavirales*[3]. Its 15 kb genomic RNA contains 10 transcriptional units encoding 11 proteins. The viral genomic RNA is tightly bound to the viral nucleoprotein N and maintained as a left-handed helical N-RNA ribonucleoprotein complex[4,5] which serves as a template for both viral RNA replication and transcription. Replication is achieved by the viral RNA-dependent RNA polymerase, which is composed of the large protein L and its main cofactor, the phosphoprotein P. Replication requires newly synthesized RNA-free N protein ($N^0$) to package the newly synthesized genomic and antigenomic RNAs (reviewed in Noton et al.[6]). RSV transcription initiates at the 3′ extragenic leader region and proceeds by a sequential stop–start mechanism that leads to the production of viral capped and polyadenylated mRNAs[6]. The transcription process requires P, L and the M2-1 protein. M2-1 is necessary to prevent premature transcription termination, both intragenically and intergenically[7–10], but its mechanism of action remains to be elucidated. M2-1 interacts with P, and RNAs[11–13], and binds with higher affinities to poly-A rich RNAs and to some gene end sequences at the 3′ end of viral mRNAs[14,15]. Based on these results Tanner et al.[15] hypothesized that M2-1 binds to viral mRNA co-transcriptionally, thereby preventing premature termination.

In infected cells, RSV induces the formation of cytoplasmic inclusions called inclusion bodies (IBs). When observed by electron microscopy, IBs appear as aggregates of amorphous or granular nature[16]. Morphologically similar cytoplasmic inclusions have been observed for a number of *Mononegavirales*[17–20]. One example are Negri bodies, cytoplasmic inclusions that are observed in rabies-infected cells, and have been used as definite histological proof of infection by this virus[21]. It has been speculated that IBs are related to aggresomes and are sites for accumulation of viral dead end products[22]. However, recent studies on rabies virus, vesicular stomatitis virus (VSV) and Ebola virus showed viral RNA synthesis occurs in IBs that can thus be regarded as viral factories[18,20,23]. Similarly, IBs observed in RSV-infected cells were shown to harbor viral proteins involved in transcription and replication of the viral genome, namely N, P, L and M2-1[24–28], as well as viral genomic RNA[26–28]. These observations led to hypothesize that IBs are the main site of RSV transcription and replication[29]. However, there is to date no direct evidence for viral RNA synthesis in IBs and the organization and function of these viral structures remain poorly characterized.

Here, we show that newly synthetized viral RNA is present in IBs of cells infected by RSV, indicating that these inclusions are sites of viral RNA synthesis. We identified functional sub-compartments within IBs that we call "IB-associated granules (IBAGs)". These IBAGs concentrate newly synthetized viral mRNA and M2-1, but exclude proteins N, P and L and the genomic RNA. We show that the formation of IBAGs is strictly dependent on RNA-dependent RNA polymerase activity. Moreover, in order to study the dynamics of IBAGs in living infected cells, we engineered a recombinant RSV expressing a functional M2-1-mGFP fusion protein. Live imaging and pulse chase analysis revealed that IBAGs are highly dynamic structures in which viral mRNA are transiently stored. Based on their protein and mRNA content and on their dynamic behavior, we propose that IBAGs are viral mRNA sorting stations in which M2-1 plays a key and hitherto undescribed role.

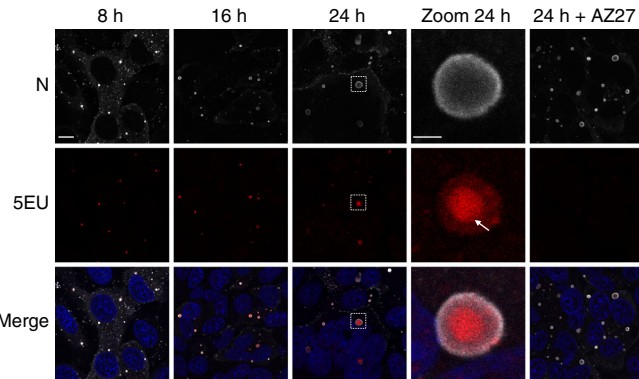

**Fig. 1** Localization of newly synthesized viral RNA in RSV-infected cells. HEp-2 cells were infected with RSV. Two hours before the indicated times p.i. cells were incubated for 1 h with actinomycin D to inhibit cellular transcription and then 5-ethynyl uridine (5EU) was added for 1 h before cells were fixed. RSV polymerase inhibitor (AZ27) was added 6 h prior actinomycin D incubation. The 5EU incorporated in newly synthesized RNAs was detected using Alexa Fluor 647-azide (*red*) and cells were stained with an anti-N antibody (*gray*) and Hoechst 33258 (merge). Representative images from three independent experiments are shown. Images were taken under a Leica SP8 confocal microscope at different times p.i. as indicated. *Scale bar* 10 μm. The boxed areas enclose IBs that are shown magnified (zoom), *scale bar* 2 μm. *White arrow* indicates a 5EU signal spot inside an IB

## Results

**Inclusion bodies are sites of viral RNA synthesis.** In infected cells, all the components of the RSV polymerase complex colocalize in cytoplasmic IBs[24–28]. In order to determine if these structures are sites of viral RNA synthesis, we performed nascent RNA staining on RSV-infected HEp-2 cells. At different times post-infection (p.i.), cells were incubated for 1 h with 5-ethynyl-uridine (5EU) in the presence of actinomycin D to inhibit cellular transcription. The incorporated 5EU was revealed and IBs were detected by immunolabeling of the N protein. IBs were detectable as early as 8 h p.i. and appeared as small spherical cytoplasmic inclusions, which became larger as the infection progressed (Fig. 1). The 5EU signal corresponding to newly synthetized viral RNAs was only visible in IBs. Noteworthy all IBs, including the largest (> 5 $\mu m^2$) exhibited strong 5EU staining (Fig. 1, panel "5EU"). When the same experiment was performed in the presence of AZ27, a specific inhibitor of RSV RNA polymerase[30,31], no 5EU staining was detected in IBs, thus confirming signal specificity (Fig. 1). These results show that IBs are sites of viral RNA synthesis.

Unexpectedly, newly synthetized viral RNAs were not homogeneously distributed within IBs, but concentrated in internal spots (Fig. 1 Zoom and Supplementary Fig. 1). This observation suggests the existence of IB sub-compartments, where newly synthesized viral RNAs concentrate.

**Viral mRNA specifically concentrate in IB-associated granules.** We next performed fluorescence in situ hybridization (FISH) experiments to determine the nature of the viral RNA concentrated in these IB subcompartments. Twenty-four hours post infection, cells were immunostained with anti-N antibodies to detect IBs, and prepared for FISH using biotinylated oligonucleotide probes designed to specifically detect either viral mRNA or genomic RNA (Table 1). FISH signal corresponding to poly (A), viral NS1 and N mRNAs was found to concentrate in spots inside IBs (Fig. 2). We named these substructures

**Table 1 Sequences of the probes used in FISH experiments**

| FISH Probe[a] | Sequence 5′–3′ biotinylated | First nucleotide position[b] |
|---|---|---|
| RSV NS1 mRNA 1 | GCTGCCCATCTCTAACCAAGGGAGTTAAATTTAAGTGGTAC | 138 |
| RSV NS1 mRNA 2 | GCCATTCAATTTGATTGTATGTATCACTGCCTTAGCCAAAGC | 287 |
| RSV NS1 mRNA 3 | TAGGGCAAATATCACTACTTGTAATAACATGCACAAACAC | 332 |
| RSV NS1 mRNA 4 | TCCATCATTTCCCATATATAACCTCCATTTTGTAGCACTGGC | 409 |
| RSV NS1 mRNA 5 | GGAGAATTTAATTTCACAATTGTCATCTATTAGACCATTAGG | 470 |
| RSV N mRNA 1 | GGAGTATCAATACTATCTCCTGTGCTCCGTTGGATGGCG | 1280 |
| RSV N mRNA 2 | CAATCAGGAGAGTCATGCCTGTATTCTGGAGCTACCTCTC | 1639 |
| RSV N mRNA 3 | TTAGCTCTCCTAATCACGGCTGTAAGACCAGATCTGTCC | 1733 |
| RSV N mRNA 4 | CTCTACTGCCACCTCTGGTAGAAGATTGTGCTATACCAAA | 1887 |
| RSV N mRNA 5 | CCGTAACATCACTTGCCCTGCACCATAGGCATTCATAAAC | 1949 |
| RSV N mRNA 6 | CAACTTGTTCCATTTCTGCTTGCACACTAGCATGTCCTAAC | 2024 |
| RSV N mRNA 7 | CTTGATTCCTCGGTGTACCTCTGTACTCTCCCATTATGCC | 2205 |
| VSV G mRNA 1 | ATTGTTCTACAGATGGAGTGAAGGATCGGATGGACTGTG | 3361 |
| VSV G mRNA 2 | GGTTGTAGAGTTATGGACAGTGGGGCATATGTAATTGCTG | 3584 |
| VSV G mRNA 3 | GAGAGGATTGGAGCAGCAATATCGACTCTGATGTATCTGG | 4108 |
| RSV genomic 1 | CACAACCCACAATGATACCACACCACAAAGACTGATGATC | 633 |
| RSV genomic 2 | CAACTTCTGTCATCCAGCAAATACGCCATCCAACGGAGC | 1186 |
| RSV genomic 3 | GCAACAAGCCAGATCAAGAACACAACCCCAACATACCTCACT | 4922 |
| RSV genomic 4 | CCACTACAAAACAACGCCAAAACAAACCACCAAACAAACC | 5124 |
| RSV genomic 5 | CTCACCGAATCAAACATTCAATGAAATCCATTGGACCTCAC | 8305 |
| RSV genomic 6 | CATCACCCAGTATCATGTATACAATGGACATCAAATATAC | 12156 |
| RSV genomic 7 | CAGCCAAATCCAACCAACTTTACACTACTACTTCCCATC | 13854 |
| RSV genomic 8 | CATCCTACACCAGAAACCCTAGAGAATATACTAGCCAATC | 13616 |
| polyA 42 | TTTTTTTTTTTTTTTTTTTTTTTTTTTTTTTTTTTTTTTTTT | |

[a]The name of the probe corresponds to the viral mRNA detected (the probe N mRNA is used to detect the N mRNA).
[b]Nucleotides are numbered according to their distance from the 5′ end of RSV and VSV virus antigenome (positive strand)

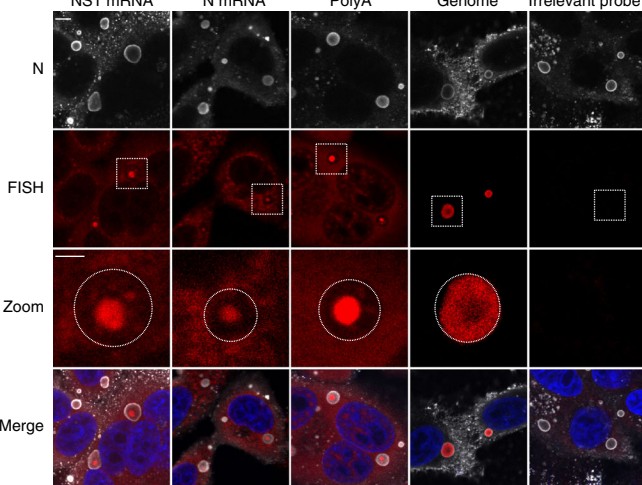

**Fig. 2** Localization of viral RNAs in RSV-infected cells. HEp-2 cells were infected with RSV for 24 h. FISH analyses were performed with specific probes (*red*) (as described in Methods) to detect polyadenylated RNA (PolyA), NS1 mRNA, N mRNA or viral genomic RNA as indicated on the pictures. A negative control was performed by probing G mRNA of VSV (irrelevant probe). Cells were also stained with an anti-N antibody (*gray*) and Hoechst 33258 (merge). Representative images from three independent experiments are shown. Images were taken under a confocal microscope. *Scale bar* 5 μm. The boxed areas enclose IBs that are shown magnified (zoom), *scale bar* 2 μm. *Circles* represent the borders of IBs

"IB-associated granules" or IBAGs. A faint FISH signal was also detected throughout the cytoplasm, suggesting the cytosolic presence of N and NS1 mRNAs. By contrast, the viral genomic RNA was detected exclusively in IBs. However, unlike viral mRNAs, the genomic RNA signal was present throughout the whole IBs except the IBAGs. The specificity of FISH signal was

verified using a probe directed against VSV G mRNA, for which no signal was observed in RSV-infected cells (Fig. 2). It is noteworthy that identical results were obtained for cells infected with an RSV isolated from a clinical specimen (Supplementary Fig. 2), ruling out a bias related to the use of a cell culture adapted strain. Together, these results reinforced the evidence of functional compartments concentrating viral mRNA in IBs.

We further analyzed the presence of IBAGs by performing poly (A) FISH staining on RSV-infected cells at different times p.i. The first IBAGs were detected as soon as 10–12 h p.i. and their number increased gradually as the viral cycle progressed (data not shown). At 24 h p.i. about 50%, of IBs as revealed by N immunostaining, exhibited one or several internal IBAGs (47 out of 85 IBs examined on 8 randomly selected fields). IBs containing visible IBAGs were significantly larger than those without (6.4 $\mu m^2$ versus 2.3 $\mu m^2$, $p < 0.001$ one-way ANOVA) (see Supplementary Fig. 3).

Having established that IBAGs concentrate RSV mRNA, we next examined the presence of cellular proteins involved in translation in IBs. Using immunostaining, we observed IB internal spots of Poly(A)binding protein (PABP) and eukaryotic translation initiation factor 4 G (eIF4G), both of which are involved in translation initiation (Fig. 3). By contrast, the small ribosomal subunit protein S6, the large ribosomal subunit protein L4 and the polypeptide release factor eRF1, a translation termination factor, were detected throughout cytoplasm but not in IBs (Fig. 3). Altogether, these results strongly suggest that despite the presence of viral mRNAs and cellular proteins of the pre-initiation complex, IBAGs are not a site of viral mRNA translation.

Cellular stress granules (SGs) also concentrate mRNA associated with translation initiation factors[32]. We therefore wanted to determine whether IBAGs are SGs included in IBs. To address this question, we stained RSV-infected cells with antibodies against the SG markers, G3BP and TIA-1, and antibodies against protein N[33]. IBAGs were visualized through

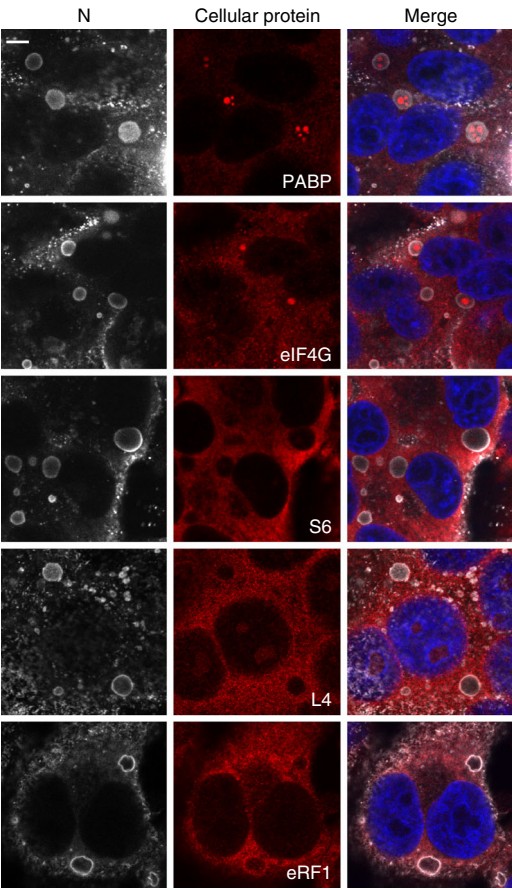

**Fig. 3** Cellular proteins required for translation do not colocalyze with IBs or IBAGs. HEp-2 cells were infected with RSV. At 24 h p.i., cells were immunostained either with an anti-PABP (mRNA interacting protein), an anti-eIF4G (translation initiation factor), an anti-S6 (small ribosomal subunit protein), an anti-L4 (large ribosomal subunit protein) or an anti-eRF1 (translation termination factor) as indicated on the pictures (cellular protein in *red*). Cells were also stained with an anti-N antibody (*gray*) to detect IBs and with Hoechst 33258 (merge). Representative images from three to four independent experiments are shown. Images were taken under a confocal microscope. *Scale bar* 5 μm

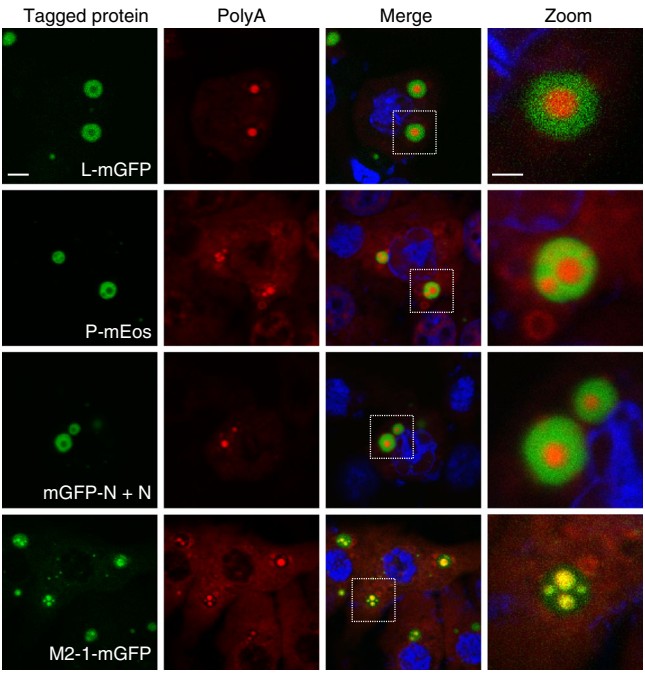

**Fig. 4** Localization of the N, P, L and M2-1 proteins inside transiently reconstituted IBs. BSRT7/5 cells were transfected with plasmids encoding the N, P, L and M2-1 proteins and the *M/Luc* subgenomic minireplicon. Tagged protein was expressed instead of the corresponding wild type as indicated on the pictures. For N protein, the plasmids encoding wild type and tagged protein were mixed in equivalent ratio when indicated mGFP-N + N. FISH analyses were performed to detect poly(A) RNA (in *red*) and were stained with Hoechst 33258 (merge). The expressed tagged proteins are visualized thanks to their spontaneous *green* fluorescence in the first column. Representative images from three independent experiments are shown. Images were taken under a Leica SP8 confocal microscope. *Scale bar* 5 μm. The boxed areas enclose IBs that are shown magnified in the fourth column, *scale bar* 2 μm

poly(A) detection. We observed SGs only in a small percentage of RSV-infected cells, consistent with previous studies[27, 34]. Furthermore, neither TIA-1 nor G3BP was detected inside IBAGs (Supplementary Fig. 4). We conclude that SGs are clearly distinct from IBs and IBAGs are not related to cellular SGs.

**M2-1 concentrates in IBAGs from which N, P and L are absent**. Since viral genomic and mRNA were found to segregate inside IBs, we next investigated the localization of N, P, L and M2-1. The main obstacle for studying the localization of proteins inside IBs is that immunostaining of N or P leads to a fluorescent signal restricted to the periphery of the IBs, presumably because of limited access to epitopes in N–P complexes[28, 35]. However, when fused to a fluorescent protein, N is observed throughout the volume of IBs[28]. For these reasons, we used fluorescent fusion proteins, rather than antibodies, to visualize RSV proteins N, P, M2-1 or L. We selected fusion proteins for which biological activity was not or only moderately affected, as determined from their ability to support transcription-replication of an RSV plasmid-based minireplicon system[36]. In this system, BSRT7/5 cells expressing T7 RNA polymerase are cotransfected with pN, pP, pM2-1 and pL expression plasmids (wild type or tagged

protein) and a vector expressing an RSV minigenome, pM/Luc, resulting in the transcription and replication of the minigenome, as revealed by reporter luciferase activity. All tagged proteins were tested independently. For protein L, we inserted the mGFP sequence in the variable region 2[37], which allowed to maintain 50% of the activity of the wild-type control (Supplementary Fig. 5). For protein P, we inserted the mEos fluorescent tag after residue 34, as this construct maintained 30% of the polymerase activity (Supplementary Fig. 5). For protein M2-1, we fused mGFP to the C-terminus of M2-1, and the resulting M2-1-mGFP exhibited the same activity as wild-type M2-1[38]. However, for the N protein, fusion of tags at either N-terminus or C-terminus did not allow to maintain any detectable polymerase activity. On the other hand, the mGFP-N fusion protein had no dominant-negative effect on RSV RNA synthesis when co-expressed with wild-type N at a ratio of 1:1 (Supplementary Fig. 5). In addition, mGFP-N was detected in both the IBs and the viral filaments at the cell membrane when expressed in RSV-infected cells, exhibiting similar localization as the wild-type N (Supplementary Fig. 6).

We transiently expressed each fusion protein, together with the other components of the minireplicon system as described above, and analyzed the localization of the different fluorescent fusion proteins in IBs by fluorescence microscopy. The pseudo-IBs that formed in these conditions were morphologically indistinguishable from those present in infected cells and similarly exhibited IBAGs, as revealed by FISH (Fig. 4). We observed that L-mGFP,

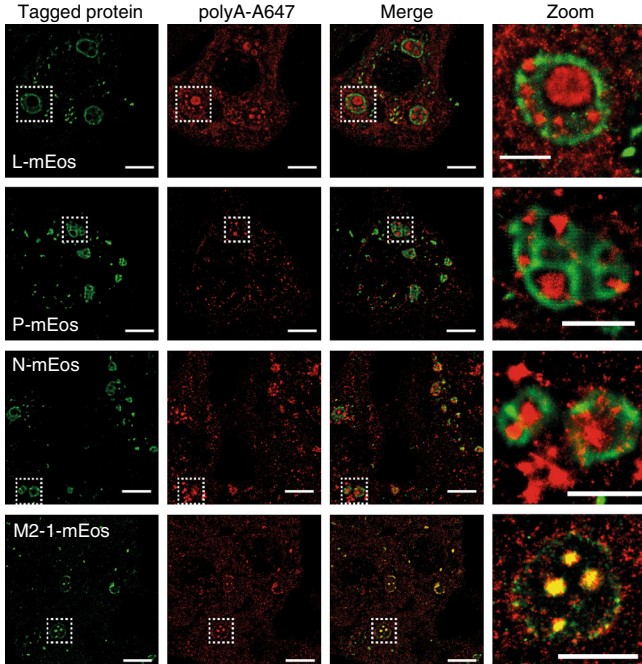

**Fig. 5** Dual color single-molecule localization microscopy of the viral proteins and mRNA inside IBs. BSRT7/5 cells were transfected with plasmids encoding the N, P, L and M2-1 proteins and the M/Luc subgenomic minireplicon. The mEos tagged protein was expressed instead of the corresponding wild-type protein as indicated on the picture. FISH analyses detecting poly(A)-mRNA were performed in order to visualize IBAGs. PALM-STORM images were realized as indicated in Methods section. The first column from the left shows a PALM image of the viral protein labeled with mEos (*green*). The second column shows a STORM image of the poly(A)-mRNA (*red*). The third column shows a superposition of the images from the first and second columns. Representative images from two independent experiments are shown, *scale bar* 5 μm. The boxed areas enclose IBs that are shown magnified in the fourth column, *scale bar* 2 μm

P-mEos and mGFP-N proteins were all present in IBs, but excluded from the IBAGs (Fig. 4). In contrast, M2-1mGFP, although still detected throughout IBs, was more concentrated in IBAGs and colocalized with mRNAs. Altogether, these results indicated that IBAGs constitute a specific area where M2-1 and viral mRNA accumulate and from which N, P and L are excluded.

**Super-resolution microscopy visualizes subdiffraction IBAGs.** The resolution of confocal microscopy is limited by diffraction to 200–300 nm, preventing visualization of IB structure at smaller scales. To characterize the organization of IBs in more details, we turned to super-resolution microscopy based on single-molecule localization (PALM/STORM), which achieves resolutions of ~20–30 nm[39, 40]. For this purpose, we used L, N and M2-1 proteins fused to the photoswitchable protein mEos, a widely used tag for PALM[41]. Each tagged protein was co-expressed with the other untagged components of the minireplicon system. The activities of the resulting mEos-tagged proteins were equivalent to that of their mGFP-tagged counterparts, as assessed with the RSV minireplicon (Supplementary Fig. 5). Staining the mRNA by FISH with probes directed against poly(A) labeled with Alexa-647 dyes, and using a buffer that promotes fluorophore blinking, enabled us to perform dual-color PALM/STORM imaging and localize the viral proteins relative to the IBAGs in pseudo-IBs (see Methods) (Fig. 5). The super-resolution images reveal in detail the architecture of the IBs. M2-1 clearly colocalize with poly(A)

spots in IBAGs while the N, P and L proteins are present all throughout the IBs volume except in the IBAGs. Moreover, they allowed us to better characterize the size distribution of IBAGs (Supplementary Fig. 7). A large fraction of IBAGs (39 out of 94, i.e. 41%) was smaller than the ~300 nm diffraction limit and their size distribution peaked near this value. The average IBAG size was 370 nm and the standard deviation 180 nm (Supplementary Fig. 7). Small clusters of mRNA were observed at or near the periphery of IBs in several PALM/STORM images. These are possibly clusters of viral mRNA being released from IBs or random clusters of cellular or viral mRNAs.

**IBAG formation depends on viral RNA synthesis.** Since M2-1 was found concentrated in IBAGs together with viral mRNA, we wondered whether IBAG formation is an active process depending on RNA-dependent RNA polymerase activity or if it reflects the inherent organization of the viral proteins. To address this, we investigated the formation of IBAGs using the mini-replicon system depleted for L, M2-1 or the subgenomic mini-genome RNA. IBs were detected using immunostaining of N, and IBAGs were revealed by FISH using a probe directed against poly (A), and by M2-1mGFP fluorescence with confocal microscopy (Fig. 6a). In the absence of pM/Luc, in the absence of pL, or upon treatment of cells with the polymerase inhibitor AZ27, M2-1mGFP displayed a homogeneous distribution inside IBs and the poly(A) signal was no longer detected in IBs (Fig. 6a). The same distribution pattern was observed when expressing the R151D M2-1mGFP impaired for RNA binding[14]. Finally, in the absence of pM2-1, poly(A) staining was no longer visible in IBs and the P-mEos signal was homogeneous, suggesting that either M2-1 itself or M2-1-dependent transcription is essential for IBAG formation (Fig. 6b). Together, these results indicate that the formation of IBAGs depends on viral mRNA synthesis.

**IBAGs are dynamic structures transiently storing viral mRNA.** We wondered whether IBAGs constitute operational sub-compartments or instead functionally irrelevant aggregates of excess products accumulating within IBs. To address this issue, we analyzed the fate of viral mRNAs and the dynamics of IBAGs in living cells. We engineered a recombinant RSV expressing a fluorescent M2-1 protein (RSV-M2-1mGFP), allowing the direct visualization of IBAGs in infected cells. Because the RSV M2 gene contains two overlapping ORFs, coding for M2-1 and M2-2 proteins[42], it is not possible to fuse mGFP to M2-1 without affecting M2-2 expression. We therefore inserted an additional cassette encoding for M2-1mGFP between the SH and G genes in the viral cDNA, and a recombinant RSV-M2-1mGFP virus expressing both M2-1 and M2-1mGFP was rescued[43] (Supplementary Fig. 8a). Single-cycle growth kinetics of RSV-M2-1mGFP was not significantly different compared with RSV (Supplementary Fig. 8b), indicating that the additional sequence of M2-1mGFP did not impact viral replication. Moreover, IBs of both wild-type RSV and RSV-M2-1mGFP developed similarly throughout viral cycle progression, indicating that M2-1mGFP does not impact IB formation (Supplementary Fig. 8c). Newly synthesized viral RNA was labeled as described above in HEp-2 cells infected with RSV-M2-1mGFP. As we observed in pseudo-IBs, M2-1mGFP was detectable throughout IBs, but concentrated in IBAGs, together with newly synthetized viral RNA (Supplementary Fig. 9a). Remarkably, M2-1mGFP and 5EU spots colocalized in all observed cells. FISH experiments performed on RSV-M2-1mGFP-infected cells confirmed that M2-1mGFP specifically concentrates in IBAGs together with viral mRNAs and that viral genomic RNA is excluded from M2-1mGFP spots (Supplementary Fig. 9b). Moreover, the

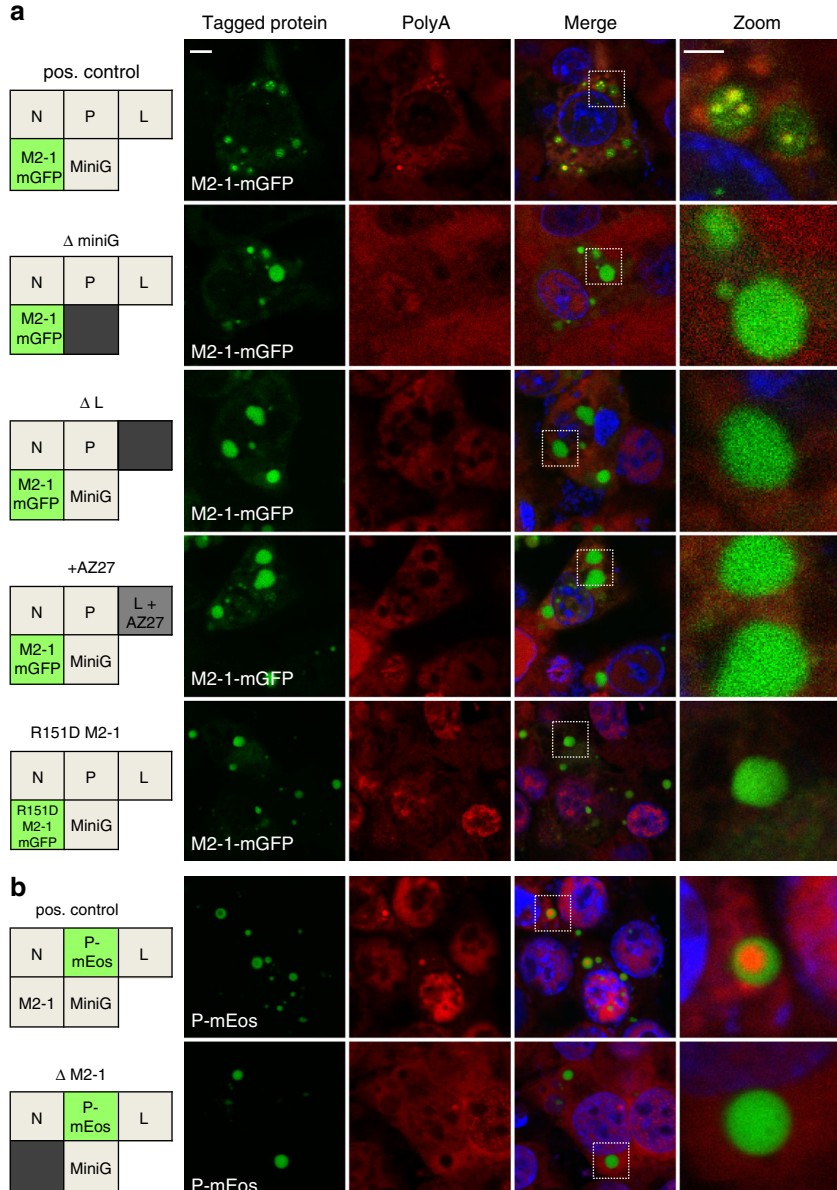

**Fig. 6** Effect of viral RNA synthesis inhibition on IBAGs formation. BSRT7/5 cells were transfected with plasmids encoding the N, P, L and M2-1 proteins and the M/Luc subgenomic minireplicon. M2-1 mGFP **a** or PmEos **b** protein was expressed instead of the corresponding wild-type protein as indicated in the boxes on the *left panel*. **a** Plasmids encoding the subgenomic minireplicon (ΔminiG) or the L protein (ΔL) were omitted, or a polymerase inhibitor (+AZ27) was added, or wild-type M2-1mGFP was replaced by the R151D M2-1mGFP (impaired in RNA binding) as indicated by a *gray box* on the *left panel*. **b** Plasmid encoding the M2-1 protein was omitted (ΔM2-1) as indicated by a *gray box* on the *left panel*. Poly(A) RNAs *(red)* were detected by FISH analyses and cells were stained with an anti-N antibody *(gray)* and Hoechst 33258 (merge). The expressed tagged proteins, visualized by epifluorescence *(green)*, are indicated in the box on the *left panel* and on the pictures. Representative images were taken under a Leica SP8 confocal microscope, *scale bar* 5 μm. The boxed areas enclose IBs that are shown magnified in the fourth column, *scale bar* 2 μm

structure of IBs and IBAGs was similar between living and fixed cells (Supplementary Fig. 10), in contrast to previous observations in primary bovine turbinate cells infected with bovine RSV[29]. These data confirmed that M2-1 specifically associates to IBAGs, and can be used as a marker of IBAGs.

We then analyzed the fate of the viral mRNAs by performing pulse chase experiments on RSV-M2-1mGFP-infected cells. Newly synthesized viral RNAs were metabolically labeled with 5EU as described above and cells were further incubated for 0, 3 or 6 h in fresh medium without 5EU (Fig. 7a). As expected, in the absence of chase (0 h), newly synthetized viral RNA was distributed throughout IBs and concentrated in IBAGs, as revealed by M2-1mGFP fluorescence (Fig. 7b). Infection was

continued in the absence of 5EU (chase) to analyze the location of newly synthesized viral RNAs 3 or 6 h later. Under these conditions, concentration of the 5EU signal in IBAGs was no longer observed, as shown in Fig. 7b. Intensity profiles shown in Fig. 7c reveal that, after 3 or 6 h of chase period, 5EU and M2-1mGFP fluorescence peaks no longer colocalized as they did directly after the pulse. The faint diffuse 5EU staining of IBs that remains after 3 and 6 h of chase may be related to 5EU incorporation in genomic and anti-genomic viral RNA (Fig. 7b, panels 3 and 6 h). Altogether, these results reveal that viral mRNAs accumulate into IBAGs together with M2-1 shortly after synthesis, but then reach a different compartment together. M2-1 might be exported together with viral mRNAs and be replaced by

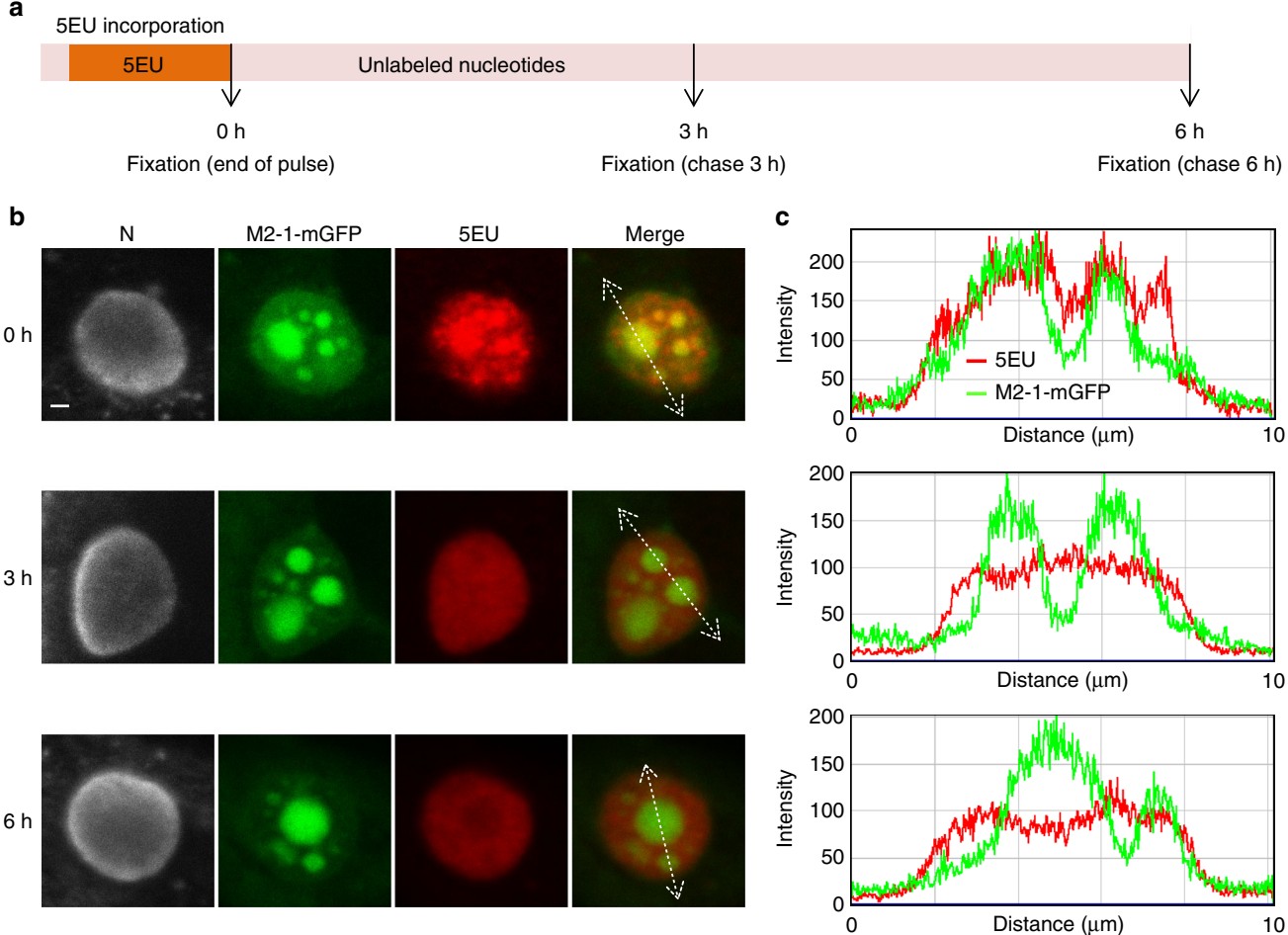

**Fig 7** Concentration of newly synthesized viral RNAs in IBAGs is transient. **a** Schematic time line of the pulse chase experiments. HEp-2 cells were infected with RSV-M2-1mGFP. At 24 h p.i. cells were incubated for 1 h with actinomycin D to inhibit cellular transcription and 5-ethynyl uridine (5EU) was added for one more hour (pulse). Cells were rinsed and fixed immediately (0 h) or after 3 or 6 h (chase). The 5EU incorporated in viral RNAs was detected using Alexa Fluor 647-azide (*red*) and cells were stained with an anti-N antibody (*gray*) and Hoechst 33258 (merge). The M2-1mGFP protein is visualized through its spontaneous green fluorescence. **b** Representative pictures from four independent experiments are shown. Images of inclusion bodies taken under a Leica SP8 confocal microscope at 0, 3 and 6 h after the 5EU pulse are presented. *Scale bar* 1 μm. The *white lines* indicate the track of a line intensity profile across the IBs **c**

novel M2-1 molecules or alternatively remain in IBAGs that accumulate newly and unlabeled viral mRNAs.

To investigate IBAG dynamics in living cells, we analyzed HEp-2 cells infected with RSV-M2-1mGFP by video-microscopy. Time-lapse fluorescence imaging of IBs in 3D (at a rate of one 30 μm image stack acquired every 3–5 min) revealed that IBAGs are highly dynamic structures. IBAGs are mobile inside IBs, grow over time and multiple IBAGs merge into spherical structures upon contact (Fig. 8a and Supplementary Movie 1). IBAGs exhibit characteristic liquid droplet behavior of membraneless intracellular RNP granules[44–46]. In large IBs exhibiting numerous IBAGs, we observed that IBAGs undergo continuous assembly–disassembly cycles, with IBAGs growing, fusing, then vanishing, as illustrated in Fig. 8a (arrow) and in Supplementary Movie 1. When large bright IBAGs split into smaller IBAGs, part of their content is released in the cytosol. This is illustrated in Fig. 8b, which shows that both the overall volume and the total fluorescence of IBs decrease over a 6-min interval, while a bright signal appears in the cytosol at close proximity of the IBs in the middle of this interval (after 3 min). Because the whole cellular volume was imaged, we can rule out the possibility that the IBAG moved out of the observed area. This observation therefore indicates that M2-1mGFP present in IBAGs is released and

diluted in the cytosol. Taken together, our data suggest that viral mRNAs concentrate in IBAGs after synthesis, and when IBAGs disassemble they are then released to the cytosol together with M2-1, which possibly acts as an mRNA carrier.

## Discussion

Previous studies have shown that RSV genome and RNA polymerase complex concentrate into cytoplasmic IBs, suggesting that these structures may host RSV transcription and replication[24–27]. We addressed this hypothesis by analyzing 5EU incorporation into nascent viral RNA, and revealed a specific 5EU staining of IBs. Our observations demonstrate that RSV IBs are a major site of viral RNA synthesis, as previously shown for *Rhabdoviridae* and *Filoviridae*[18–20, 23]. It is noteworthy that IBs are not clearly visible before 8 h p.i., and that the metabolic labeling of newly synthesized viral RNA is inconclusive at earlier time points. We assume that, at early times, viral RNA synthesis is carried out either within some IBs too small to be detected, or in the cytosol, as observed for VSV[20]. Lifland et al. detected no RSV genomic RNA in IBs larger than 5 μm[228]. In contrast, we found that all IBs, including the larger ones, contain newly synthetized viral RNAs and genomic RNA. It is established that the

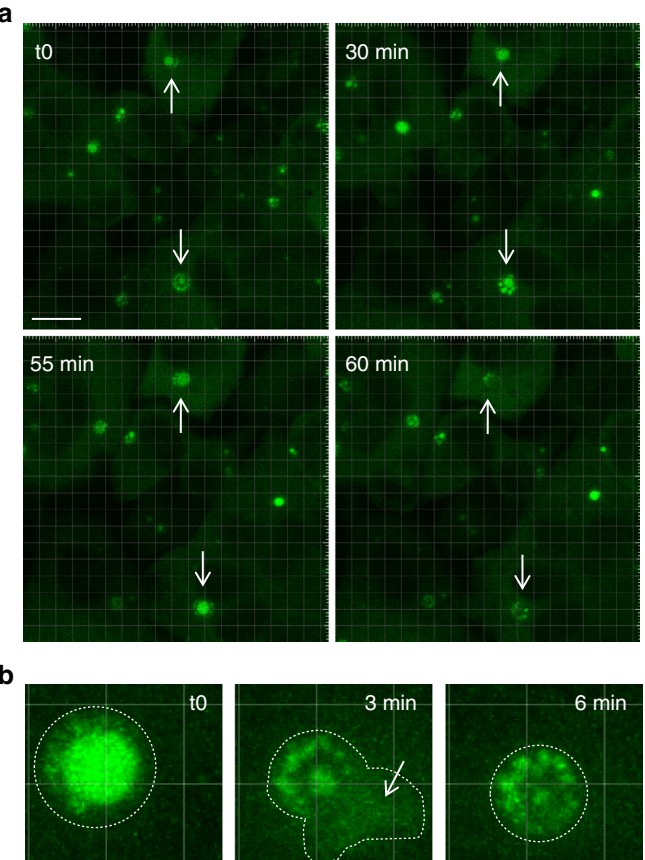

**Fig 8** IBAGs are dynamic structures. Time-lapse microscopy of IBAGs in HEp-2 cells infected with RSV-M2-1mGFP. At 24 h p.i., cells were imaged in a chamber heated at 37 °C, with a Leica SP8 confocal microscope. The M2-1mGFP protein was visualized by *green* fluorescence. Representative images from five independent experiments are shown. **a** Representative images of IBAG dynamics. *White arrows* indicate IBs undergoing IBAG growth, followed by IBAG collapse. *Scale bar* 10 μm. **b** Time-lapse imaging of an IB showing IBAG disassembly (regions of interest). The *dotted line* indicates the IB boundary as determined on the first and last image. The central image suggests the release of the IBAG content into the cytoplasm indicated by a *white arrow*. *Scale bar* 1 μm

viral genomic RNA is poorly accessible because it is tightly associated with the N protein[47]. This poor accessibility may impair its detection in live cells as performed by Lifland et al. We detected viral RNA by metabolic labeling, which is likely unaffected by RNA accessibility, or by FISH after performing denaturation steps on fixed cells, which may expose the viral RNA to solvent. The differences in staining protocols may thus explain the different results. Altogether, our observations support the hypothesis that all IBs (even the largest ones) are competent for RNA synthesis as observed for rhabdoviruses[18, 20].

Strikingly, we showed that nascent viral RNA, but not genomic RNA, concentrates inside IB sub-compartments that we termed IBAGs. We used a transient transfection system to reconstitute pseudo-IBs in conditions supporting the transcription/replication of an RSV minigenome, and selectively tagged the N, P, L or M2-1 proteins by fluorescent protein fusions. This system allowed us to visualize by confocal and PALM/STORM super-resolution microscopy that M2-1 concentrates in IBAGs together with viral mRNA, while N, P and L are excluded from them. Exclusion of N from IBAGs is consistent with the exclusion of the viral genome, as suggested by FISH performed on infected cells. It was

previously determined that the P-binding and RNA-binding domains partially overlap on the surface of M2-1 and that P and RNA bind to M2-1 in a competitive manner[14, 15]. Thus, the observed segregation of P and IBAGs, where M2-1 and mRNAs co-localize, is consistent with these data.

Our data show that IBAGs formation is strictly dependent on viral RNA synthesis. IBAGs appear early during the course of infection, the first ones being visible from 12 h p.i., after which their number increases as the viral cycle progresses. However, IBAGs are not visible in all IBs. This might be due to the dynamic behavior of IBAGs or the detection limits of light microscopy, rather than to functional differences between IBs. Indeed, live imaging revealed that IBAGs undergo continuous cycles of assembly and disassembly. Moreover, our PALM/STORM data showed that IBAGs have a relatively broad distribution of sizes above and below the resolution of a confocal microscope (~250 nm), revealing the presence of numerous IBAGs with sizes < ~300 nm in diameter. It is thus possible that many IBAGs are too small to be detected, especially in small IBs. To our knowledge, this is the first description of sub-compartments inside IBs induced by RSV infection. The small size of IBAGs, combined with the inadequacy of immunostaining within IBs[28], may explain why IBAGs have not been described previously. This analysis paves the way for further super-resolution characterization of functional IB organization in the case of RSV, but also for other *Mononegavirales*. For example, *Filoviridae* express VP30, a transcription factor that shares functional and structural similarities with M2-1[14]. In the case of *Rhabdoviridae*, when analyzing relationships between IBs and SGs in cells infected by Rabies, Nikolic et al.[48] revealed small dots of newly synthetized viral RNA inside IBs that resemble the IBAGs identified here.

In eukaryotic cells, non-translating mRNAs stalled in translation initiation often assemble with proteins into visible cytoplasmic structures without membrane, termed mRNP granules or RNA granules. Over the last few decades, these granules have emerged as an important class of membrane-less organelles, shown to be implicated in crucial aspects of mRNA regulation, including control of mRNA localization, translation and stability[32, 49, 50]. Our results indicate that IBAGs present similarities to mRNP granules, since they (i) transiently concentrate untranslated viral mRNAs with translation initiation factors (PABP and eIF4G) and (ii) exhibit liquid droplet behavior as reported for cellular mRNP granules[45, 51]. However, IBAGs are different from cellular mRNP granules, since they do not exhibit cellular SG markers (G3BP and TIA-1) and contain specific viral material: viral mRNA and viral M2-1 protein. IBAGs can thus be regarded as viral-specific mRNP granules. Like cellular mRNP granules, IBAGs could be involved in the sorting of viral mRNAs and/or in regulation of translation and stability of viral mRNAs. According to the current model, cellular mRNP granules can be regarded as a liquid–liquid phase separation driven by multiple low-affinity interactions[44, 46, 51–53]. This phenomenon occurs when a molecule or a mixture of molecules forms a network of weak interactions leading to concentration of the molecules in a separate phase. Thus it is possible that IBAGs emerge when enough mRNA protein complexes concentrate into IBs to form immiscible droplets that undergo a phase transition. This hypothesis is consistent with the disappearance of IBAGs when viral RNA synthesis is abolished. Because M2-1 forms tetramers, it has the ability to bind several mRNAs simultaneously and might thus promote the formation of interaction networks[36]. The inability of mutant M2-1 (R151D), impaired for RNA binding, to promote IBAG formation supports this hypothesis. However, since this mutation also impairs viral transcription we cannot rule out an indirect effect[14].

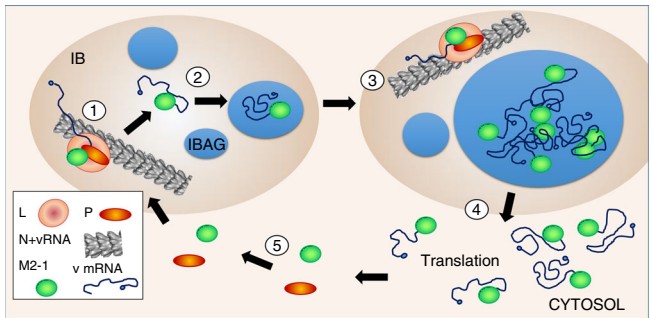

**Fig 9** Model of viral mRNA trafficking. (1) Viral mRNA synthesis occurs in IBs in area containing genomic viral RNA and N, L, P and M2-1 (designated as peripheral area). M2-1 binds to nascent mRNA poly(A) tail. (2) These RNA–protein complex together with cellular proteins mediate condensation of IBAGs recruiting the newly synthetized mRNA-M2-1 complexes. (3) IBAGs grow and fuse together. (4) When reaching critical size, IBAGs disassembly releases the mRNAs most probably still bound to M2-1 into the cytoplasm. (5) Newly synthetized M2-1 (and possibly M2-1 released from viral mRNA) bind with P in the cytoplasm and is imported in the peripheral area of IBs

Based on our data, we propose the following model for viral mRNA and M2-1 trafficking (Fig. 9). The exclusion of the N, P and L proteins, as well as the viral genomic RNA, from the IBAGs strongly suggests that viral transcription and replication do not take place in these structures, but most probably occur in other areas of IBs. Then, shortly after synthesis, viral mRNAs accumulate in IBAGs, leading to (i) the almost exclusive localization of IB viral mRNA inside IBAGs and (ii) the concentration of newly synthetized viral RNA in IBAGs. M2-1 was proposed to bind nascent viral mRNA since it preferentially binds to poly(A) sequences[15]. Our data suggest that M2-1 remains bound to the released nascent mRNAs as they concentrate into IBAGs. As demonstrated by our pulse chase experiments, newly synthesized viral mRNAs only transit through IBAGs, since IBAGs are not their final destination. Moreover, ribosomal proteins (S6 and L4) and eRF1 translation termination factor are absent from IBs, suggesting that no translation occurs in these areas. These results are in agreement with previous findings about VSV, showing that cytoplasmic export of viral mRNAs is necessary for their translation[20]. In this context, we speculate that viral mRNAs bound to M2-1 are released from IBAGs toward the cytosol, where they likely undergo translation. Live imaging revealed that IBAGs disassemble when they reach a certain size and/or density, releasing their components into the cytosol. However, the underlying mechanisms are unknown. The observation by super-resolution microscopy of small mRNA heaps at the periphery of IBs suggests that mRNAs are released as small RNA granules rather than as individual mRNAs.

Until now, M 2-1 was described as a viral transcription anti-terminator, which prevents premature transcription termination both intragenically and intergenically[7–10]. Previous publications suggest that the M2-1 transcription anti-terminator function could be linked to its binding to nascent viral mRNA[14, 15]. Our data show that M2-1 remains bound to viral mRNA in IBAGs and probably also after their export in the cytosol. In conclusion, it is likely that M2-1 is also involved in the post-transcriptional metabolism of viral mRNA, highlighting a new function for this protein.

## Methods

**Cells and viruses**. HEp-2 cells (ATCC number CCL-23) were grown in Eagle's minimum essential medium (MEM) and BHK-21 cells (clone BSRT7/5) constitutively expressing the T7 RNA polymerase[54] were grown in Dulbecco modified essential medium (DMEM). Media were supplemented with 10% (v/v) fetal calf serum (FCS) and antibiotics. The wild-type RSV and the RSV-M2-1mGFP are derived from the RSV subtype A Long strain (ATCC VR-26) and were rescued by reverse genetics as previously described[43]. Viruses were amplified on HEp-2 cells as recommended by ATCC at 33 °C. Experiments were performed with viral stock amplified after three to five passages. The RSV (Group A) isolated from a clinical specimen on HEp-2 cells at 37 °C was passaged twice on HEp-2 cells. Plaque assay were performed at 37 °C on HEp-2 cells using Avicel overlay as previously described[43].

**Reagents and antibodies**. Actinomycin D, dextran sulfate and streptavidin were from Sigma. 5EU was from Invitrogen. Formaldehyde was from Polysciences. AZ27 was obtained from AstraZeneca R&D Boston[30, 31]. Stock solutions of actinomycin D (1 mg/ml), streptavidin (1 mg/ml), dextran sulfate (40%, w/v) and 5EU (100 mM) were prepared in water. Formaldehyde (4%, v/v) was prepared in phosphate-buffered saline (PBS). AZ27 1 mM was stored in dimethyl sulfoxide and working dilution was prepared at 100 μM in MEM FCS 2% extemporaneously. The rabbit polyclonal anti-N was obtained by repeated injection of purified recombinant protein produced in *Escherichia coli* as described in ref. [55]. The mouse anticellular proteins antibodies: PABP (10E10; 1 μg/ml), eIF4G (A-10; 1 μg/ml), S6 (C-6; 1 μg/ml), L4 (RQ-7; 2 μg/ml) were from Santa Cruz. The rabbit anti-eRF1 (GTX 108271; 2 μg/ml) was from Gene Tex, the goat anti-TIA-1 (C-20; 2 μg/ml) was from Santa Cruz and the mouse anti-G3BP (2F3; 5 μg/ml) was from Sigma. Secondary antibodies (2 μg/ml) raised against mouse, rabbit or goat IgG (H + L) and conjugated to Alexa Fluor 488, 594 or 647 were from Invitrogen.

**Plasmids**. All the viral sequences were derived from the human RSV strain Long, ATCC VR-26 (GenBank accession AY911262.1). Expression plasmids of RSV N, P, P-eGFP, L, eGFP-L and M2-1 proteins designated as pN, pP,pP-eGFP, pL, p-eGFP-L and pM2-1 have been previously described[36, 37, 56]. The pP-mEos and pL-mEos were derived from pP-eGFP and peGFP-L expression vectors by replacing the eGPF coding sequence by the mEos2[41] coding sequence using respectively the *Bam*HI and the *Sal*I sites. The pmGFP-N and pmEos-N were constructed by adding a *Bam*HI site upstream of the N coding sequence by site-directed mutagenesis (Stratagene) in pN and inserting either the mGFP coding sequence (Clontech) or the mEos2 coding sequence in frame with N open reading frame (ORF) in the *Bam*HI site. The stop codon of the M2-1 ORF in pM2-1 was deleted by site-directed mutagenesis (Stratagene) and either the mGFP coding sequence (Clontech) or the mEos2 coding sequence was inserted in *Xho*I restriction site in frame with M2-1 ORF to obtain the p-M2-1mGFP and pM2-1mEos. The pM/Luc subgenomic replicon, which contains the firefly luciferase (Luc) gene under the control of the M/SH gene start sequence, has been described previously[36]. The full-length cDNA clone of RSV pACNR-rHRSV (GenBank accession KF713490) was previously described[43]. The M2-1mGFP coding sequence was amplified from the pM2-1mGFP vector by PCR using specific primers (Supplementary Table 1) containing RSV gene start and gene end sequences from the RSV SH gene and was cloned into a *Eco*RV restriction site between the RSV SH and G genes in the pACNR-rHRSV. All constructs were verified by sequencing. The nucleotide sequence of RSV-M2-1mGFP was deposited in the Genbank nucleotide database with accession code KX348546.

**Cell infection, transfection and minigenome assay**. Infections were performed at 37 °C on exponentially growing HEp-2 cells at a multiplicity of infection of 1. BSR/T7-5 at 90% confluence in 24-wells plates were transfected with Lipofectamine 2000 (Invitrogen) according to the manufacturer recommendations with a plasmid mixture containing 125 ng of pN, 125 ng of pP, 62.5 ng of pL, 31 ng of pM2-1 and 125 ng of pM/Luc, as well as 31 ng of pβ-Gal (expressing β-galactosidase gene under CMV promoter control; Promega) to normalize transfection efficiencies. To inhibit L polymerase activity, AZ27 was added at 1 μM all along transfection. At 24 h post-transfection cells were either fixed for immunostaining or lysed in luciferase lysis buffer (30mM Tris pH 7.9, 10 mM MgCl₂, 1 mM DTT, 1% Triton X-100 and 15% glycerol). Luciferase activity expressed in RLU was measured using a Tecan infinite M200PRO plate reader and normalized on β-galactosidase activity.

**Incorporation of ethynyluridine in nascent viral RNA**. Cells were grown and infected on glass coverslips in 24-wells plates. At different times p.i., medium was replaced by 250 μl per well of complete medium FCS 2% (v/v) supplemented with actinomycin D (20 μg/ml) to inhibit cellular transcription. One hour later, 250 μl of the same medium supplemented with 5EU (2 mM) were added to the wells containing actinomycin D for one more hour (5EU incorporation in viral RNA). To inhibit L polymerase activity, AZ27 was added at 1 μM during 6 h before actinomycin D incubation. Next, cells were rinsed with PBS and fixed with PBS-formaldehyde 4% (v/v) for 10 min at 4 °C, washed with PBS and permeabilized with PBS-bovine serum albumin (BSA) 1% (w/v)-Triton X-100 0.1% (v/v) for 10 min. 5EU labeling was then performed according to the manufacturer's instructions (Invitrogen, Click-iT RNA Imaging Kit). Finally, immunofluorescence staining was performed as described below. For 5EU pulse-chase experiments, incorporation and labeling of 5EU was performed as described above. After a pulse of 1 h, cells were rinsed with fresh complete medium and fixed with PBS-formaldehyde 4% (v/v) at 0, 3 or 6 h post-incorporation.

**Immunofluorescence staining**. Cells were grown on glass coverslips in 24 well plates. After infection and treatments, cells were fixed with PBS-formaldehyde 4% (v/v) for 10 min at 4 °C, washed with PBS and permeabilized with PBS-BSA 1% (w/v)-Triton X-100 0.1% (v/v) for 10 min. Cells were incubated for 1 h in PBS-BSA 1% (w/v) with the appropriate primary antibodies, following the manufacturer's recommendations. Cells were rinsed with PBS, incubated for 30 min with the appropriate Alexa Fluor-conjugated secondary antibodies. Nuclei were stained with Hoechst 33342 (1 µg/ml) for 5 min and cells were washed with PBS. Coverslips were then mounted in ProLong gold antifade reagent (Thermofisher). Cells were examined by confocal microscopy under the WLL Leica SP8 microscope and representative pictures were taken. When combined with other staining (FISH, 5EU incorporation), immunofluorescence was performed as the last step. For the labeling of the anti-L4 ribosomal large subunit protein, cells were fixed with a 70% EtOH/H$_2$O (v/v) fixation during 1 min at −20 °C.

**Fluorescent in situ hybridization**. Cells were infected, fixed and permeabilized as described above. Endogenous biotin was blocked in PBS-BSA 1% (w/v) supplemented with free streptavidin (4 µg/ml) for 1 h. Coverslips were rinsed three times with PBS, post-fixed 10 min at 4 °C in formaldehyde 4% (v/v), rinsed two times with PBS and incubated in hybridization mix (2× SSC (1× SSC is 150 mM NaCl and 15 mM sodium citrate), dextran 10% (w/v), formamide 20% or 50% (v/v) depending on probes that were used, 1 mg/ml herring sperm DNA and 1 µM of total hybridization probes), in a humidified chamber at 37 °C for 3 h. All probes were single-stranded DNA oligonucleotides with a 3′-biotinylation (Sigma). Total mRNAs (Poly(A)) were detected by using one probe (Poly(dT)) whereas specific viral or cellular mRNAs were detected by using a mix of several probes (see below). Next, cells were washed two times at 42 °C with the following three solutions: 2× SSC plus formamide 20% or 50% (v/v) depending on probes that were used, 2× SSC, 1× SSC and finally PBS at room temperature. Probes were then detected by incubating cells with streptavidin-Alexa Fluor 647 conjugate (8 µg/ml) in PBS-BSA 1% (w/v) during 1 h prior to three washes with PBS. Cells were then submitted to immunofluorescent staining and confocal microscopy. Twenty percent formamide was used during hybridization and rinsing when comparing Poly(A) and negative control VSV staining, whereas 50% formamide was used when comparing all other probes with VSV negative control. Sequences of probes for in situ hybridization are given in Table 1.

**Imaging of IBs and internal structures**. Confocal microscopy was used to study characteristics of individual IBs. Z-stack image acquisitions of multi-labeled (Hoechst, N, M2-1mGFP, FISH or 5EU) cells were performed under a ×63 apochromatic lens and a numerical zoom comprised between ×1 and ×5 (LSAF acquisition software). For quantifications, eight independent stacks at zoom ×2.5 of each experimental condition were taken, allowing a detailed analysis of approximately 40 cells, corresponding to 85 IBs. Image analyses were realized with the Image J software. The number and size of IBs (based on N staining) were determined by using the ImageJ's "analyze particles" function. Specific labeling of FISH Poly(A) inside IBs was used to determine the number of IBAGs per IBs. The same parameters were applied for all experimental conditions and quality of image segmentation was checked by comparing object staining with their calculated outlines. For time lapse microscopy, HEp-2 cells were seeded on Ibidi µ-slide eight-well polymer coverslips and infected with RSV-M2-1mGFP as described above. At 24 h p.i., cells were placed under a Leica SP8 confocal microscope equipped with a chamber heated at 37 °C. Z-stacks about 30 µm thick with a 0.3 µm Z-step were acquired every 5 min during 4 h. Videos were finally visualized in 3D using the Imaris software.

**Dual COLOR PALM/STORM microscopy**. The super-resolution microscopy experiments were done on a custom-built microscope as described in more detail before[57]. This microscope features a Nikon Ti-E Eclipse body equipped with a perfect focus system, which ensures that the same z-section remains in focus throughout PALM/STORM imaging, a ×100, NA = 1.45 objective lens and an ultrasensitive EMCCD camera (IXON-DV887ECS-BV, Andor, Belfast, Northern Ireland). The attached laser box is composed of four high power lasers: a 405 nm–100 mW laser from Oxxius to activate photoswitching of mEos, a 488 nm–500 mW laser from MPB Communications for excitation of unactivated mEos, a 561 nm–500 mW laser from MPB Communications to excite activated mEos and a 642 nm–500 mW laser from MPB Communications to excite the fluorescent molecule A647.

The cells were grown on coverslips and prepared as previously explained. Just before imaging, an oxygen scavenger buffer, composed of glucose oxidase, catalase, glucose and MEA, was prepared (see ref. [58]). In order to correct for lateral drift during image sequence acquisition and to register localizations from the two color channels, 100 nm fluorescent multicolor Tetraspec beads were added to the sample. The round coverslip was fixed on the coverglass, which was thermally coated with a parafilm. A square hole fitting the coverslip was cut in the parafilm, and filled with the oxygen scavenger buffer. The coverslip was placed on top of this hole, with the cells directly in contact with the solution. The sample was then placed on the microscope and the perfect focus turned on. When a suitable field of view was found, we sequentially acquired images of A647 first and mEos second. Because of the high density of poly(A), 30,000–50,000 raw images of A647 were acquired, while 20,000 images were sufficient to deplete mEos fluorescence. The parameters of the camera

were 50 ms exposure time and 300 EM gain for A647, and 100 ms exposure time and 300 EM gain for mEos, respectively. For A647 imaging, only the 642 nm laser was used. For mEos, the 488 nm laser was used for initial widefield imaging, then both the 405 and 561 nm lasers were used for activation and excitation, respectively. The 405 nm laser power was continuously increased in order to maintain the number of activated (and not yet photobleached) molecules roughly constant.

A modified version of the Matlab based software MTT[59] was used to compute molecular localizations in each channel and an in-house Matlab software (PALMvis) to correct for drifts and chromatic shifts using images of Tetraspec beads, and for super-resolution image visualizations. Based on the standard deviation of coordinates in small localization clusters, the resolution of the PALM/STORM images was estimated to ~40–50 nm or better.

**Data availability**. The nucleotide sequence of RSV-M2-1mGFP is available in the Genbank nucleotide database with accession code KX348546. The authors declare that all other data supporting the findings of this study are available within the article and its Supplementary Information files, or are available from the authors upon request.

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

## Acknowledgements

We thank Dr Qin Yu from AstraZeneca R&D Boston—USA, for providing the AZ27 drug. We are grateful to Felix Rey, Jean-Louis Gaillard, Charles-Adrien Richard and Monica Bajorek for helpful discussions. Benoit Maury is gratefully acknowledged for technical assistance in microscopy experiments. We are grateful to the Cymages platform for access to SP8 Leica, which was supported by grants from the region Ile-de-France. We acknowledge Ignacio Izeddin for the gift of a mEos2 encoding plasmid. We acknowledge support from the INSERM and the Versailles Saint-Quentin University. This study was supported by grants from the ANR (French national research agency), RESPISYNCY-CELL project, from the Region Ile-de-France and by the "Assistance Publique des Hôpitaux de Paris" (AP-HP). M.L and C.Z. acknowledge support from Institut Pasteur and Région Ile de France (DIM Malinf).

## Author contributions

V.R., M.L., J.F.E., C.Z. and M.A.R.W. conceived and designed the experiments. V.R., M.L., C.B. and M.A.R.W. performed the experiments. V.R., M.L., D.S., S.B., M.G., J.F.E., E.G., C.Z. and M.A.R.W. analyzed the data. V.R., M.L., D.S., M.G., J.F.E., C.Z., E.G. and M.A.R.W. wrote the paper.

## Additional information

**Competing interests:** The authors declare no competing financial interests.

