## [Peer Review File · Nature Communications]

Reviewers' comments:

Reviewer #1 (Remarks to the Author):

The authors described and characterized the spatial and functional organization of inclusion bodies (IB) in Respiratory Syncytial Virus (RSV) infected cells. The major finding is the identification of a novel subcompartment within the inclusion bodies which is named IB associated granules (IBAGS). It is shown that newly synthesized viral mRNA concentrated in IBAGS, whereas genomic RNA is mainly present at the periphery of IB. While poly(A)binding protein and the translation initiation factor eIF4G were detected in IBAGS the ribosomal S6 and L4 proteins and the polypeptide release factor eRF1 were absent suggesting that IBAGS are not the site of viral mRNA translation. The absence of TIA-1 and G3BP, both markers of stress granules, led the authors to the conclusion that IBAGS are not related to cellular stress granules. Interestingly, confocal microscopy revealed that the nucleocapsid proteins L, P and N while present in IB are excluded from IBAGS. M2-1, however, a transcription processivity factor of RSV accumulated in IBAGS. FISH experiments detected viral m-RNA in M2-1mGFP spots where genomic RNA was excluded. This observation was confirmed by PALM/STORM super-resolution microscopy. Using a minireplicon system for RSV the authors showed, that IBAGS formation is dependent on viral RNA synthesis and the presence of M2-1. Live cell microscopy and pulse chase experiments revealed IBAGS to be highly dynamic structures, which seem to release their content periodically into the cytosol.

In conclusion, the authors suggest, that viral mRNA is concentrated in IBAGS and, when IBAGS disassemble, are exported together with M2-1 into the cytosol for translation. These experiments highlight a new function for M2-1 that, in addition to its role in viral transcription, might have a function in mRNA stabilization and transport.

General comments

The data presented in this manuscript are convincing, novel and highly interesting also for readers outside virology. This work will stimulate the discussion of the role of viral cytoplasmic inclusion bodies that has been somehow overlooked in the past.

The methods used in the manuscript are up-to-date and the experiments have been comprehensively described. In addition, the outline of the manuscript makes it easy to follow the line of arguments which leads to the well supported conclusion on the role of the newly detected and characterized inclusion body subcompartment in RSV-infected cells.

Specific comments

Line 139 Typo: M3-1 change to M2-1

Line 196: The term "evolutionary pattern" does not seem to be appropriate here. May be "...IBs of both...developed similarly..."

Line 210: better use "newly synthesized"

Line 213: Consider using "directly after the pulse"

Line 243 Typo: ARN change to RNA

Reviewer #2 (Remarks to the Author):

RSV-induced inclusion bodies (IBs) are multi-functional bodies, thought to be involved both in viral RNA synthesis and in antagonizing innate immunity. A substantial body of work has indicated that proteins associated with the viral polymerase complex (N, P, L, and M2-1), as well as viral genomic RNA, concentrate in IBs. However, direct viral synthesis has not been demonstrated. Rincheval et al. use a variety of recently developed RNA and protein labeling techniques to show that viral mRNAs are synthesized in IBs and accumulate at distinct sites (termed IBAGs) within IBs, in contrast to viral

genomic RNA which is present throughout. They also show that of the viral polymerase-associated proteins, only M2-1 co-localizes with mRNA in IBAGs, and that IBAGs are distinct from stress granules and are not sites of translation. Together with the previously published findings that M2-1 can bind poly A sequences, this is strong evidence of an as-of-yet unknown role for M2-1 in the synthesis, stability, storage or transport of viral mRNAs.

The study is in general well-written and rigorously executed and contributes substantial and novel insights into the mechanisms of RSV mRNA synthesis. The findings are important to the RSV field. The main claims described above are adequately supported by the data. A few issues need clarification:

1) Genome-detecting probes show absence of RSV genome in IBAGs. Probes specific for mRNA and polyA prove that mRNA is present in IBAGs. The mRNA probes can however also anneal to the viral anti-genome, and thus it cannot be excluded that a portion of the signal detected with mRNA probes is due to anti-genome presence in IBAGs. This reviewer is not familiar with the quantitative contribution of anti-genome to viral replication. However, the potential of the probes to detect anti-genome in addition to mRNA needs to be addressed. In retrospect, it would have been useful if some of the probes to detect mRNA sense were designed to anneal to the intergenic regions as negative controls for anti-genome.

2) Limitation of fusion proteins for protein localization: The fact that GFP-N does not interfere with N in a mini-replicon assay (Fig. S4) does not prove that GFP-N trafficking behavior accurately reflects N trafficking. The GFP component of GFP-N may preclude participation of N in certain functions and in targeting to IBAGs, especially since N-GFP does not support RNA replication. Because of this, the statement that N does not target to IBAGs is a bit premature even though it would be a logical conclusion as mRNAs are not encapsidated. This concern does not negatively impact the main findings which concern M2-1.

A related note:

Authors say that N antibodies restrict the signal to the periphery. However, since several primary (PAPB, eIF4G) as well as conjugated secondary antibodies can penetrate IBs and IBAGs (Fig 3, 5) this is not likely due to the IB but more likely due to the N Ab itself or the fact that structurally distinct populations of N may exist in the center and periphery of IBs. Different antibodies, or maybe different detergents, might be able to allow more accurate or more complete visualization of N than GFP-N. Have the authors tried to confirm their findings with other N antibodies or an epitope-tagged N protein? This would increase confidence in the data related to N trafficking.

3) Fig 5: The poly A signal is rather different between the zoomed-in images, with the top right panel having most of its signal in a large central IBAG whereas the lower right panel (M2-1) has its poly-A signal mostly in small peripheral sites. This creates the impression that some of the Eos fusion proteins may affect localization of mRNAs more than others, or that the images are not representative of the main phenotype?

4) Line 96 (Fig 2): They say that viral genome is more abundant on the periphery of IBs. The evidence for this is not convincing.

5) It is stated in the results section (line 235) that data strongly suggest that IBAGs are exported together with M2-1. This may be the case, however no evidence is presented to support this.

6) Line 106: add statistical validation for claiming 'significant' differences between IBs with or without IBAGs.

7) Line 97 and 647 say VSV M mRNA probes were used, whereas Table 1 says that VSV G mRNA

probes were used. Please clarify.

8) Prior findings to be considered for discussion:

- In a live cell characterization of bovine RSV, Santangelo et al (2006) showed that IBs in living cells have a structure that is quite different than the IB structure seen in fixed cells. These findings are relevant to the current observations and should be discussed.
- Lifland et al (2012) suggests that different roles may be played in RSV genome replication by small versus large IBs. Since the authors have examined their findings in differently sized IBs, it would be helpful if the authors discussed these prior findings in light of their own findings.
- A paper showing eIF2a phosphorylation by RSV (Lindquist et al, 2011) is relevant to the findings of Fig. 3, and could be included and discussed.
- M2-1 is known to be important in viral RNA replication as an anti-terminator. The authors proposed a very different function for M2-1 relating to mRNA. Can the authors discuss how M2-1 could carry out two distinct roles?

Reviewer #3 (Remarks to the Author):

Rincheval and colleagues in this submission describe an innovative approach to better understand the role of the respiratory syncytial virus M2-1 protein in virus replication. It has been known for some time that RSV replication complexes are located in cytosolic inclusion bodies, as are the replication machineries of several other members of the mononegavirales. Major findings of this study are that M2-1 concentrates viral mRNAs in granular substructures within the inclusion bodies. These IBAGs subsequently disintegrate and release the viral mRNAs into the cytosol for translation and viral protein synthesis.

These findings advance the field by providing important novel insight into the sorting mechanism of RSV genomic and messenger RNAs. This previously unappreciated role of the M2-1 protein adds to the fundamental understanding of RSV replication and may open novel avenues for therapeutic interference with RSV protein expression. Experiments shown are mostly of high quality and well interpreted, but several specific points need to be addressed to fully substantiate the authors' conclusions.

Specific points:

1) The single molecule microscopy study in figure 5 summarizes an excellent approach to test for localization of RSV polymerase complex proteins and mRNAs in the IB substructures. However, while the formation of local aggregates can be reasonably appreciated in the PolyA-A647 stains of the upper three panels (testing colocalization with L, P, or N, respectively), the staining pattern of the most important fourth panel (testing for colocalization with M2-1) is markedly different and does not show any clearly appreciable centers of local aggregation. What is the reason for the stark difference in the PolyA pattern seen, for instance, in the fourth panel of figure 4 and what has motivated the selection of the boxed region in the "merge" section of figure 5? Surprisingly, data shown in figure S4 suggest equal bioactivities of both M2-1 constructs despite the phenotypic differences seen in figures 4 and 5. Since the data shown in figure 5 are central to the study's conclusions, these discrepancies need to be addressed.

2) Testing the significance of the minireplicon-based findings in virus infected cells is important and the recombinant RSV-M2-1mGFP close is nicely designed. However, it remains unclear whether this virus was engineered to express exclusively M2-1mGFP and M2, or expresses both tagged M2-1mGFP and standard M2-1. If the latter, little can be concluded from the growth curve shown in figure S6b. Please clarify.

3) Some members of the author team have described a number of interesting mutant M2-1 proteins in previous work (Blondot et al, PPath 8, 2012) that are selectively impaired in interaction with P or viral

RNA, respectively. Including these M2-1 mutants in a minireplicon study as shown in figure 6 would be much more informative than leaving out entire polymerase complex components or pharmaceutically blocking all RNA synthesis.

4) Are viral mRNAs really actively exported from IBAGs (lines 307-308)? To me, it rather appears that individual IBAGs collapse and release contents in the cytosol when they reach a certain size and/or density. If you believe active transportation occurs, please substantiate. Otherwise, reword. There are several other occasions where you seem to apply an active process without confirmation (i.e. line 296 "...the exclusion..."). Exclusion implicates an active process (keeping the other components out), while most likely M2-1:mRNAs simply have some tendency to spontaneously aggregate. Please check the manuscript in its entirety and reword misleading statements.

Minor edits:

-line 28: "...is the primary cause of..."

-line 139: should be M2-1, unless you discovered a completely new RSV protein

Reviewer #4 (Remarks to the Author):

"Functional organization of cytoplasmic inclusion bodies in cells infected by Respiratory Syncytial Virus" by Rincheval et al. elucidates the organization and function of cytoplasmic inclusion Bodies (IBs) in respiratory syncytial virus (RSV) infections. I have been asked to comment specifically on the superresolution microscopy component of the manuscript, so I limit my points below to only the microscopy components of the paper and not the biological interpretation. As a note from a non-expert reader though, the text uses an excess of abbreviations, which makes it less accessible than I'd expect for Nature Communications.

Major concern:

*Figure 5: My main concern about this superresolution figure is that one important conclusion here should be that there is a lot of N, L, and P outside of the IBs. This is a typical thing that superresolution can find: due to higher sensitivity, smaller populations that are invisible to confocal can be found with a superresolution microscope. Compare Figures 4 and 5. However, this is not addressed at all in the paper. With a good analysis, the authors could compare the % of N, L, and P in the IBs to that outside.

*Given the amount of protein outside of the cell, I'm concerned about the strength of the conclusion that "The PALM/STORM images also revealed the presence of mRNA clusters at or near the periphery of IBs (Fig. 5), suggesting the export of clustered mRNAs from IBs." Especially since this conclusion is based on just a few images. At the very least, something like stats such as percentage of cells with clusters outside the IBs would be helpful.

*Or, is this an artifact related to the limited function of the fusion proteins? I'm very concerned that conclusions are being made based on fusion proteins to L and P that maintain only 30 - 50% of their wild type activity levels. And if fusions to N are inactive, how do the authors know that the localization of N-GFP is representative of the wild type protein localization in the 1:1 mixture?

More minor points:

Figure 1: "Unexpectedly, newly synthesized viral RNAs were not homogeneously distributed within IBs, but concentrated in internal spots (Fig. 1 Zoom). This observation suggests the existence of IB sub-compartments, where newly synthesized viral RNAs concentrate." It's not obvious from the Fig. 1 Zoom images that the RNA is not evenly dispersed throughout the IB. This conclusion requires some

sort of control – something that is smooth in the confocal microscope vs. 5EU here forming punctate spots.

Or are the authors just referring to the fact that the RNA is more concentrated at the center of the IB than the edges? Might this not just be an artifact of imaging a 3D IB, so of course there's more RNA at the center because the IB is thicker at the center? This could be examined in the confocal z stacks that were acquired according to materials and methods

Figure 2: Here, (unlike Figure 1), the concentrated spots (named IBAGs by the authors) are observable, but again here, at confocal resolution, statements like "genomic RNA signal was present in the whole IBs but being more abundant at the periphery" are hard to defend. (Also "being" here is grammatically incorrect).

Page 6: "The main obstacle for studying the localization of proteins inside IBs is that immunostaining of N or P leads to a fluorescent signal restricted to the periphery of the IBs..." how have the authors ascertained that this sort of artifact is not also a problem for RNA labeling?

Figure 5: The difference between N, L, and P localization vs M21 localization is impressive. A sentence in the text pointing this out would add weight to the claims that there is exclusion of N, L, P from IBAGs.

Figure 5: The authors should mention in the transition from Figure 4 to Figure 5 that the PALM/STORM experiments in STORM buffer are no longer live-cell experiments (vs. Figure 4).

Reviewer #1 (Remarks to the Author):

The authors described and characterized the spatial and functional organization of inclusion bodies (IB) in Respiratory Syncytial Virus (RSV) infected cells. The major finding is the identification of a novel subcompartment within the inclusion bodies which is named IB associated granules (IBAGS). It is shown that newly synthesized viral mRNA concentrated in IBAGS, whereas genomic RNA is mainly present at the periphery of IB. While poly(A)binding protein and the translation initiation factor eIF4G were detected in IBAGS the ribosomal S6 and L4 proteins and the polypeptide release factor eRF1 were absent suggesting that IBAGS are not the site of viral mRNA translation. The absence of TIA-1 and G3BP, both markers of stress granules, led the authors to the conclusion that IBAGS are not related to cellular stress granules. Interestingly, confocal microscopy revealed that the nucleocapsid proteins L, P and N while present in IB are excluded from IBAGS. M2-1, however, a transcription processivity factor of RSV accumulated in IBAGS. FISH experiments detected viral m-RNA in M2-1mGFP spots where genomic RNA was excluded. This observation was confirmed by PALM/STORM super-resolution microscopy. Using a minireplicon system for RSV the authors showed, that IBAGS formation is dependent on viral RNA synthesis and the presence of M2-1. Live cell microscopy and pulse chase experiments revealed IBAGS to be highly dynamic structures, which seem to release their content periodically into the cytosol. In conclusion, the authors suggest, that viral mRNA is concentrated in IBAGS and, when IBAGS disassemble, are exported together with M2-1 into the cytosol for translation. These experiments highlight a new function for M2-1 that, in addition to its role in viral transcription, might have a function in mRNA stabilization and transport.

General comments

The data presented in this manuscript are convincing, novel and highly interesting also for readers outside virology. This work will stimulate the discussion of the role of viral cytoplasmic inclusion bodies that has been somehow overlooked in the past. The methods used in the manuscript are up-to-date and the experiments have been comprehensively described. In addition, the outline of the manuscript makes it easy to follow the line of arguments which leads to the well supported conclusion on the role of the newly detected and characterized inclusion body subcompartment in RSV-infected cells.

We thank the reviewer for this positive appreciation of our work.

Specific comments

We thank the reviewer for noticing these typos and the wrong phrasing, which have been corrected in the revised version of the manuscript

Line 139 Typo: M3-1 change to M2-1

This error has been corrected

Line 196: The term "evolutionary pattern" does not seem to be appropriate here. May be "...IBs of both...developed similarly..."

The sentence "Moreover, IBs of both wild type RSV and RSV-M2-1mGFP **exhibited the same evolutionary pattern** throughout viral cycle progression..." was changed to "Moreover, IBs of both wild type RSV and RSV-M2-1mGFP **developed similarly** throughout viral cycle progression..."

Line 210: better use "newly synthesized"

The term "**neo-synthesized**" has been replaced by "**newly synthesized**".

Line 213: Consider using "directly after the pulse"

"**in the absence of chase**" was replaced by "**directly after the pulse**"

Line 243 Typo: ARN change to RNA

Done

Reviewer #2 (Remarks to the Author):

RSV-induced inclusion bodies (IBs) are multi-functional bodies, thought to be involved both in viral RNA synthesis and in antagonizing innate immunity. A substantial body of work has indicated that proteins associated with the viral polymerase complex (N, P, L, and M2-1), as well as viral genomic RNA, concentrate in IBs. However, direct viral synthesis has not been demonstrated. Rincheval et al. use a variety of recently developed RNA and protein labeling techniques to show that viral mRNAs are synthesized in IBs and accumulate at distinct sites (termed IBAGs) within IBs, in contrast to viral genomic RNA which is present throughout. They also show that of the viral polymerase-associated proteins, only M2-1 co-localizes with mRNA in IBAGs, and that IBAGs are distinct from stress granules and are not sites of translation. Together with the previously published findings that M2-1 can bind poly A sequences, this is strong evidence of an as-of-yet unknown role for M2-1 in the synthesis, stability, storage or transport of viral mRNAs.

The study is in general well-written and rigorously executed and contributes substantial and novel insights into the mechanisms of RSV mRNA synthesis. The findings are important to the RSV field. The main claims described above are adequately supported by the data. A few issues need clarification:

We thank the reviewer for this positive appreciation of our work.

1) Genome-detecting probes show absence of RSV genome in IBAGs. Probes specific for mRNA and polyA prove that mRNA is present in IBAGs. The mRNA probes can however also anneal to the viral anti-genome, and thus it cannot be excluded that a portion of the signal detected with mRNA probes is due to anti-genome presence in IBAGs. This reviewer is not familiar with the quantitative contribution of anti-genome to viral replication. However, the potential of the probes to detect anti-genome in addition to mRNA needs to be addressed.

We thank the reviewer for this insightful comment. We agree that the probes directed against N or NS1 mRNA could in theory also reveal the antigenomic RNA. However, it is unlikely that the signal observed in IBAGs is due to the presence of the antigenomic RNA because (1) the poly(dT) probe which does not recognize the antigenomic viral RNA, gives a signal similar to that of probes specific to mRNA, (2) the antigenomic RNA is encapsidated by N, which is excluded from IBAGs and (3) it is now accepted that the antigenomic RNA is much less abundant than the mRNAs. Indeed, Bermingham et al. previously performed Northern blots on infected cell lysates using probes specific to both antigenome and mRNA. The signal corresponding to the antigenome was much fainter than the signal corresponding to the monocistronic mRNA and the polycistronic readthrough mRNAs detected with the probe (Bermingham et al, 1999 PMC18021) (see also our response to the next comment below).

In retrospect, it would have been useful if some of the probes to detect mRNA sense were designed to anneal to the intergenic regions as negative controls for anti-genome.

The positive sense, intergenic regions are not fully specific of antigenomic RNA. Transcription termination of RSV polymerase is not 100% efficient and leads to synthesis of polycistronic readthrough mRNAs. These polycistronic mRNAs, which contain intergenic regions, are at least as abundant as the antigenome (Bermingham et al, 1999 PMC18021). Therefore, it would not be possible to discriminate between antigenome or viral mRNAs by using probes directed against intergenic regions. Moreover, detection of genomic RNA by FISH is difficult, due to poor accessibility of the genomic RNA, which is encapsidated by N during the entire viral replication cycle. The antigenomic RNA is most likely very hard to detect because it is much less abundant than the genomic RNA.

2) Limitation of fusion proteins for protein localization: The fact that GFP-N does not interfere with N in a mini-replicon assay (Fig. S4) does not prove that GFP-N trafficking behavior accurately reflects N trafficking. The GFP component of GFP-N may preclude participation of N in certain functions and in targeting to IBAGs, especially since N-GFP does not support RNA replication. Because of this, the statement that N does not target to IBAGs is a bit premature even though it would be a logical conclusion as mRNAs are not encapsidated. This concern does not negatively impact the main findings which concern M2-1.

In order to address this concern, we infected cells transiently expressing a fluorescent N protein (mGFP-N) with wild type RSV and compared the locations of mGFP-N protein and WT N. The distribution of mGFP-N and WT N stained by antibodies was very similar. We detected the fluorescent mGFP-N both in the IBs (except IBAGs) and the viral filaments at the cell membrane. The latter observation shows that mGFP-N is incorporated into the viral ribonucleocapsid and viral particles. This is consistent with the ability of recombinant mGFP-N to encapsidate RNA, forming ring-like structures like WT N, as previously shown by Roux et al. (Roux et al, 2008 PMID: 18335041).

Moreover, as highlighted by the reviewer, the data obtained using mGFP-N are consistent with the localization of the viral genomic RNA and of the P protein (which is known to bind N). Taken together, these data strongly suggest that the locations of mGFP-N and WT N are identical.

To clarify this point, we added a Supplementary Figure (Supplementary Fig.6.) and modified the following sentence in the results section:

“**However**, the mGFP-N fusion protein had no dominant negative effect on RSV RNA synthesis when **expressed together** with wild type N at a ratio of 1:1 (Supplementary Fig. 4)” was replaced by:

“**On the other hand**, the mGFP-N fusion protein had no dominant negative effect on RSV RNA synthesis when **co-expressed** with wild type N at a ratio of 1:1 (Supplementary Fig. 5). **In addition, mGFP-N was detected in both the IBs and the viral filaments at the cell membrane when expressed in RSV infected cells, exhibiting similar localization as the wild type N. (supplementary Fig. 6)”**

A related note:

Authors say that N antibodies restrict the signal to the periphery. However, since several primary (PAPB, eIF4G) as well as conjugated secondary antibodies can penetrate IBs and IBAGs (Fig 3, 5) this is not likely due to the IB but more likely due to the N Ab itself or the fact that structurally distinct populations of N may exist in the center and periphery of IBs.

We acknowledge that this is a valid concern. The ring-like staining of IBs with anti-N antibodies can potentially result from a lack of penetration of antibodies into IBs. However, as highlighted by the reviewer, this would be in contradiction with the ability to immunostain cellular proteins inside IBs. Other authors also detected cellular proteins inside IBs using antibodies or viral RNA by FISH (Lifland et al. 2012 PMID: 22623778; Lindquist et al. 2010 PMID: 20844027; Brown et al. 2005 PMID: 15936795; Utley et al. 2008 PMID: 18621683; Radhakrishnan et al. 2010 PMID: 20530633; Fricke et al 2013 PMID: 23152511). We hypothesize that, in IBs, N and P are engaged in a tight interaction network, that hides some epitopes, thus explaining the lack of immunodetection of these proteins.

When cells were fixed in formaldehyde 4% for 24h then shortly permeabilized with methanol the N staining pattern looks like the mGFP-N one (see below). We did not use these conditions to avoid any alteration of IBs structures by prolonged fixation and methanol treatment.

We did not include this image in the revised manuscript in order not to distract from the central points of our work.

Different antibodies, or maybe different detergents, might be able to allow more accurate or more complete visualization of N than GFP-N. Have the authors tried to confirm their findings with other N antibodies or an epitope-tagged N protein? This would increase confidence in the data related to N trafficking.

It is unlikely that this ring pattern depends on the antibodies used since it has been observed by different authors using different antibodies against N (monoclonal anti-N B023 (Lifland et al. 2012 PMID: 22623778); monoclonal anti-N B130 (Utley et al. 2008 PMID: 18621683), polyclonal rabbit serum in the present work.). Moreover, the same observations were done using anti-P antibodies (Lindquist et al. 2010 PMID: 20844027; Fricke et al 2013 PMID: 23152511).

3) Fig 5: The poly A signal is rather different between the zoomed-in images, with the top right panel having most of its signal in a large central IBAG whereas the lower right panel (M2-1) has its poly-A signal mostly in small peripheral sites. This creates the impression that some of the Eos fusion proteins may affect localization of mRNAs more than others, or that the images are not representative of the main phenotype?

We acknowledge this concern. The differences between the IBAG patterns that appeared in Fig 5 were most likely due to poor transfection efficiencies in these particular experiments. We have now repeated the experiment with newly purified vectors and low passage cells in two independent experiments. The polyA staining in these new experiments is now similar to the pattern obtained with the other tagged proteins. The PmEos transfection was repeated in parallel as a control. We replaced the images of PmEos and M2-1mEos in Figure 5 based on these new experiments.

4) Line 96 (Fig 2): They say that viral genome is more abundant on the periphery of IBs. The evidence for this is not convincing.

The genomic staining pattern strongly contrasts with the mRNA staining. The staining intensity at the periphery is clearly stronger than in the center. We provided a different image that shows this more clearly and updated Figure 2 accordingly. The complete exclusion of N from the IBAGs is consistent with the absence of genomic RNA in this area.

5) It is stated in the results section (line 235) that data strongly suggest that IBAGs are exported together with M2-1. This may be the case, however no evidence is presented to support this.

We agree with the reviewer that we have no evidence for an active process. Thus, we replaced “are then **exported** to the cytosol together with M2-1” by “are then **released** to the cytosol together with M2-1”.

We believe that our data suggests, but does not demonstrate, that the mRNA are released when IBAGs disassemble, because:

1. We observe that M2-1 signal decreases inside IBs and concomitantly increases in the cytosol strongly suggesting the release of M2-1 in the cytosol

2. The pulse chase experiments clearly show that viral mRNAs do not remain permanently in IBAGs
3. The cytosol, where translation occurs, is the most likely final destination of viral mRNAs
4. M2-1 is known to bind viral mRNA and both co-localize in IBAGs.

We thus replaced “our data **strongly** suggest” by “our data suggest”.

- 6) Line 106: add statistical validation for claiming ‘significant’ differences between IBs with or without IBAGs.

We added the p value of our statistical test: “(6.4 μm^2 versus 2.3 μm^2 , **p<0,001 one way ANOVA**) (see supplementary Fig. 2).” For better readability, we presented the differences between IBs with and without IBAGs in Supplementary Fig. 2 and the complete statistical data are provided in the legend.

- 7) Line 97 and 647 say VSV M mRNA probes were used, whereas Table 1 says that VSV G mRNA probes were used. Please clarify.

We thank the reviewer for pointing out this error. The probe is directed against VSV G. The text has been corrected accordingly.

- 8) Prior findings to be considered for discussion:

- In a live cell characterization of bovine RSV, Santangelo et al (2006) showed that IBs in living cells have a structure that is quite different than the IB structure seen in fixed cells. These findings are relevant to the current observations and should be discussed.

We thank the reviewer for this interesting remark. Our results regarding this point are clear. We observed no differences in IB structure between living and fixed cells, as shown by M2-1 mGFP imaging. To support this point, we added **Supplementary Fig.10** and the following sentence in the results section **“Moreover the structure of IBs and IBAGs were similar between living and fixed cells (Supplementary Fig.10), in contrast to previous observations in primary bovine turbinate cells infected with bovine RSV (Santangelo et al, 2006).”**

An in-depth discussion of the differences between our results and those of Santangelo et al. would be difficult, since the virus, the cells, the time post-infection and the methods of detection (detection of viral RNA in living cells using dual-labeled oligonucleotide probes with a reporter fluorophore at one end and a quencher at the other) are different.

- Lifland et al (2012) suggests that different roles may be played in RSV genome replication by small versus large IBs. Since the authors have examined their findings in differently sized IBs, it would be helpful if the authors discussed these prior findings in light of their own findings.

We thank the reviewer for this comment. We have added a discussion of this point in the Discussion section as follows:

“Lifland et al. detected no RSV genomic RNA in IBs larger than 5 μm^2 (Lifland et al. 2012). In contrast, we found that all IBs, including the larger ones, contain newly synthesized viral RNAs and genomic RNA. It is established that the viral genomic RNA is poorly accessible because it is tightly associated with the N protein (Tawar et al 2009). This poor accessibility

may impair its detection in live cells as performed by Lifland et al. We detected viral RNA by metabolic labelling likely unaffected by RNA accessibility, or by FISH after denaturation steps on fixed cells, which may expose viral RNA to solvent. The differences in staining protocols may thus explain the different results. Altogether, our observations support the hypothesis that all IBs (even the largest ones) are competent for RNA synthesis as observed for rhabdoviruses (Lahaye et al. 2009; Heinrich et al. 2010).”

Previous version:

“We also showed that all IBs, including large IBs (area > 5µm²) observed after 24h p.i., contain newly synthesized viral RNAs. This last observation supports the hypothesis that all IBs (even the largest ones) are competent for RNA synthesis.”

- A paper showing eIF2a phosphorylation by RSV (Lindquist et al, 2011) is relevant to the findings of Fig. 3, and could be included and discussed.

The work by Lindquist et al addresses the formation of stress granules in RSV infected cells, but not eIF2a phosphorylation itself. The aim of Figure 3 was to investigate whether translation of viral mRNA can occur in IBAGs by detecting transcription initiation factors like eIF4G. We believe that it is not appropriate to discuss the phosphorylation of eIF2a, since it is not directly related to the focus of our study.

- M2-1 is known to be important in viral RNA replication as an anti-terminator. The authors proposed a very different function for M2-1 relating to mRNA. Can the authors discuss how M2-1 could carry out two distinct roles?

At this point, we actually do not explain the antiterminator function of M2-1 at the molecular level. It has been proposed that this “antiterminator” function of M2-1 is linked to its capacity to bind to viral mRNA, in particular Gene End and poly-A sequences which are present at the 3’ end of viral mRNA (Tanner et al, 2014, PMID:24434552; Blondot et al, 2012, PMID: 22675274). The possible binding of M2-1 to viral mRNAs after transcription highlights another function of M2-1 that also relates to its RNA binding properties.

Accordingly, we modified the last paragraph of the Discussion section as follows:

“ Until now, M2-1 was described as a viral transcription anti terminator, which prevents premature transcription termination both intra- and intergenically⁷⁻¹⁰. **Previous publications suggest that the M2-1 transcription anti terminator function could be linked to its binding to nascent viral mRNA^{14,15}**. Our data show that M2-1 remains **bound to** viral mRNA in IBAGs and probably **also** after **their** export in the cytosol. **In conclusion, it is likely** that M2-1 is also involved in the post-transcriptional metabolism of viral mRNA, highlighting a new function for this protein. »

Instead of:

“ Until now, M2-1 was described as a viral transcription anti terminator, which prevents premature transcription termination both intra-and intergenically.⁷⁻¹⁰. Our data show that M2-1 remains associated with fully transcribed viral mRNA in IBAGs and most

probably after export in the cytoplasm. This suggests that M2-1 is also involved in the post-transcriptional metabolism of viral mRNA, highlighting a new function for this protein. »

Reviewer #3 (Remarks to the Author):

Rincheval and colleagues in this submission describe an innovative approach to better understand the role of the respiratory syncytial virus M2-1 protein in virus replication. It has been known for some time that RSV replication complexes are located in cytosolic inclusion bodies, as are the replication machineries of several other members of the mononegavirales. Major findings of this study are that M2-1 concentrates viral mRNAs in granular substructures within the inclusion bodies. These IBAGs subsequently disintegrate and release the viral mRNAs into the cytosol for translation and viral protein synthesis. These findings advance the field by providing important novel insight into the sorting mechanism of RSV genomic and messenger RNAs. This previously unappreciated role of the M2-1 protein adds to the fundamental understanding of RSV replication and may open novel avenues for therapeutic interference with RSV protein expression. Experiments shown are mostly of high quality and well interpreted, but several specific points need to be addressed to fully substantiate the authors' conclusions.

We thank the reviewer for this positive appreciation of our work.

Specific points:

1) The single molecule microscopy study in figure 5 summarizes an excellent approach to test for localization of RSV polymerase complex proteins and mRNAs in the IB substructures. However, while the formation of local aggregates can be reasonably appreciated in the PolyA-A647 stains of the upper three panels (testing colocalization with L, P, or N, respectively), the staining pattern of the most important fourth panel (testing for colocalization with M2-1) is markedly different and does not show any clearly appreciable centers of local aggregation. What is the reason for the stark difference in the PolyA pattern seen, for instance, in the fourth panel of figure 4 and what has motivated the selection of the boxed region in the “merge” section of figure 5? Surprisingly, data shown in figure S4 suggest equal bioactivities of both M2-1 constructs despite the phenotypic differences seen in figures 4 and 5. Since the data shown in figure 5 are central to the study's conclusions, these discrepancies need to be addressed.

We agree with this comment. We repeated the experiment and obtained clearer and more convincing results. Figure 5 was modified accordingly. Please see the response to reviewer 2 point 3.

2) Testing the significance of the minireplicon-based findings in virus infected cells is important and the recombinant RSV-M2-1mGFP clone is nicely designed. However, it remains unclear whether this virus was engineered to express exclusively M2-1mGFP and M2, or expresses both tagged M2-1mGFP and standard M2-1. If the latter, little can be concluded from the growth curve shown in figure S6b. Please clarify.

The RSV-M2-1mGFP virus expresses both standard M2-1 and M2-1mGFP. The text has been modified to clarify this point (last sub section of the results first paragraph):

“a recombinant RSV-M2-1mGFP virus **expressing both M2-1 and M2-1mGFP** was rescued” instead of

“a recombinant RSV-M2-1mGFP virus was rescued”.

Supplementary figure 6a (renamed 8a in the revised version) shows a schematic representation of the RSV-M2-1mGFP infectious clone. The color of M2-1 (blank) was modified to highlight the two M2-1 genes. We think that the growth curves are relevant since they show that the additional M2-1mGFP does not impair viral growth in cell culture, as stated in the Results: “Single-cycle growth kinetics of RSV-M2-1mGFP were not significantly different compared with RSV (Supplementary Fig. 6b), indicating that the additional sequence of M2-1mGFP did not impact viral replication.”

3) Some members of the author team have described a number of interesting mutant M2-1 proteins in previous work (Blondot et al, PPath 8, 2012) that are selectively impaired in interaction with P or viral RNA, respectively. Including these M2-1 mutants in a minireplicon study as shown in figure 6 would be much more informative than leaving out entire polymerase complex components or pharmaceutically blocking all RNA synthesis.

We thank the reviewer for this suggestion, which helped us to significantly improve our manuscript. We tested a mutant M2-1 affected in RNA binding. As expected, the mutated M2-1mGFP localizes throughout IBs, but IBAGs are no longer observed, neither from the M2-1mGFP signal nor from polyA staining). These data have been added to Figure 6 and commented in the Results and Discussion sections, as follows:

In the results section (lines 186-187) we added:

“The same distribution pattern was observed when expressing the R151D M2-1mGFP impaired for its interaction with RNA (Blondot et al, 2012).”

The following sentence was added in the discussion section (lines 309-310)

“The inability of mutant M2-1 (R151D), impaired for RNA binding, to promote IBAG formation supports this hypothesis. However, since this mutation also impairs viral transcription, we cannot rule out an indirect effect (Blondot et al, 2012). »

The mutant protein M2-1 impaired for its interaction with P is no longer recruited to IBs, as we previously showed in Blondot et al, 2012.

4) Are viral mRNAs really actively exported from IBAGs (lines 307-308)? To me, it rather appears that individual IBAGs collapse and release contents in the cytosol when they reach a certain size and/or density. If you believe active transportation occurs, please substantiate. Otherwise, reword.

There are several other occasions where you seem to apply an active process without confirmation (i.e. line 296 “...the exclusion...”). Exclusion implicates an active process (keeping the other components out), while most likely M2-1:mRNAs simply have some tendency to spontaneously aggregate. Please check the manuscript in its entirety and reword misleading statements.

We agree that we do not have any proof of an active export of the mRNA and that our use of the term “export” might have been misleading. In order to avoid any misunderstanding, we therefore reworded the appropriate passages as follows.

Lines 324-326, we replaced the following sentences:

“In this context, we speculate that viral mRNAs bound to M2-1 are exported from IBAGs toward the cytosol, where they likely undergo translation. Live imaging revealed that IBAGs disassemble, releasing their components into the cytosol.”

by :

“In this context, we speculate that viral mRNAs bound to M2-1 are **released** from IBAGs toward the cytosol, where they likely undergo translation. Live imaging revealed that IBAGs disassemble **when they reach a certain size and/or density**, releasing their components into the cytosol.”

Line 174: we replaced :

“suggesting the export of clustered mRNAs from IBs.”

by:

“suggesting the **release** of clustered mRNAs from IBs.”

Line 328 : we replaced:

“that mRNAs are exported as small RNA granules rather than as individual mRNAs”

by:

“that mRNAs are **released** as small RNA granules rather than as individual mRNAs”

Line 225: we replaced:

“but are then exported to a different compartment”

by:

“but **reach** then a different compartment.”

Regarding our usage of the term “exclusion”: we simply meant to say that proteins N, P and L and the genomic viral RNA are absent from IBAGs. We believe that “exclusion” is appropriate in this context and do not share the reviewer’s view that this term necessarily implies an active process. For example, water and oil exclude each other when mixed, but this process does not require addition of energy to the system. We therefore chose to keep this terminology in the paper.

Minor edits:

-line 28: “...is the primary cause of...”

-line 139: should be M2-1, unless you discovered a completely new RSV protein

We thank the reviewer for pointing out these mistakes, which we corrected:

-Line 28: we replaced:

“Human respiratory syncytial virus (RSV) is the first cause”

by

“Human respiratory syncytial virus (RSV) is the primary cause”

-Line 139: M3-1 has been replaced by M2-1

Reviewer #4 (Remarks to the Author):

“Functional organization of cytoplasmic inclusion bodies in cells infected by Respiratory Syncytial Virus” by Rincheval et al. elucidates the organization and function of cytoplasmic inclusion Bodies (IBs) in respiratory syncytial virus (RSV) infections. I have been asked to comment specifically on the superresolution microscopy component of the manuscript, so I limit my points below to only the microscopy components of the paper and not the biological interpretation.

As a note from a non-expert reader though, the text uses an excess of abbreviations, which makes it less accessible than I’d expect for Nature Communications.

We modified the text to replace unnecessary abbreviations (which were used less than 3 times).

MNV was replaced by Mononegavirales

RdRp was replaced by “RNA-dependent RNA polymerase”

SGs was replaced by “cellular stress granules” in the Discussion (in the Results section it is used in only one paragraph where the abbreviation is defined)

“FISH using a poly(dT) probe” was replaced by “ FISH using a probe directed against poly(A)”

Major concern:

*Figure 5: My main concern about this superresolution figure is that one important conclusion here should be that there is a lot of N, L, and P outside of the IBs. This is a typical thing that superresolution can find: due to higher sensitivity, smaller populations that are invisible to confocal can be found with a superresolution microscope. Compare Figures 4 and 5. However, this is not addressed at all in the paper. With a good analysis, the authors could compare the % of N, L, and P in the IBs to that outside.

We agree that analyzing the presence of N, P or L outside the IBs in infected cells could be interesting. However, for technical reasons such as the use of multiple fluorescent proteins, this is currently not feasible with infected cells. Note that the presence of viral proteins outside of IBs is consistent with the hypothesis that they are translated in the cytosol. The PALM/STORM experiments presented here were performed with cells transiently transfected with 5 plasmids to express the viral proteins and a minigenome. These conditions enable the reconstitution of functional IBs to study their structures. However, in this artificial system viral proteins are over-expressed, which can induce their aggregation in the cytosol and make the quantifications suggested by the reviewer of little biological relevance.

*Given the amount of protein outside of the cell, I’m concerned about the strength of the conclusion that “The PALM/STORM images also revealed the presence of mRNA clusters at or near the periphery of IBs (Fig. 5), suggesting the export of clustered mRNAs from IBs.” Especially since this conclusion is based on just a few images. At the very least, something like stats such as percentage of cells with clusters outside the IBs would be helpful.

We agree that our conclusion was too strong. Note that the strong polyA staining in the cells is expected since poly(dT) probes reveal not only viral but also cellular mRNAs. Our purpose was to highlight the presence of mRNA clusters at the periphery of the IBs in several images to mention the possibility that viral mRNA might perhaps cluster even outside IBAGs.

In response to this comment, we modified the sentence:

“The PALM/STORM images also revealed the presence of mRNA clusters at or near the periphery of IBs (Fig. 5), suggesting the export of clustered mRNAs from IBs.”

as follows:

“Small clusters of mRNA were observed at or near the periphery of IBs in several PALM/STORM images. These are possibly clusters of viral mRNA being released from IBs or random clusters of cellular or viral mRNAs.”

*Or, is this an artifact related to the limited function of the fusion proteins? I’m very concerned that conclusions are being made based on fusion proteins to L and P that maintain only 30 – 50% of their wild type activity levels. And if fusions to N are inactive, how do the authors know that the localization of N-GFP is representative of the wild type protein localization in the 1:1 mixture?

Reviewer 2 raised the same issue (point 2). We provide an additional supplementary figure and modified the text. Please see response to reviewer 2 (point 2).

We acknowledge that the biological activities of L and P are diminished as compared to wild type proteins, but these fusion proteins nevertheless maintain a substantial activity. The L-mGFP activity was sufficient to allow the rescue of a viable recombinant RSV expressing L-mGFP instead of wild type L (recent unpublished data from our lab). This virus seems to grow very well, as it reaches the same titer as wild type RSV, namely about 2×10^6 PFU/mL. Moreover, the localization observed for all the viral proteins and the different viral RNAs (genomic and mRNA) are consistent with their biological functions and known interactions regardless of the tagged protein or the model (infection or transfection). We therefore consider it valid to assume that the localizations of fluorescent proteins reflect those of wild type proteins.

More minor points:

Figure 1: “Unexpectedly, newly synthesized viral RNAs were not homogeneously distributed within IBs, but concentrated in internal spots (Fig. 1 Zoom). This observation suggests the existence of IB sub-compartments, where newly synthesized viral RNAs concentrate.” It’s not obvious from the Fig. 1 Zoom images that the RNA is not evenly dispersed throughout the IB. This conclusion requires some sort of control – something that is smooth in the confocal microscope vs. 5EU here forming punctate spots.

Or are the authors just referring to the fact that the RNA is more concentrated at the center of the IB than the edges? Might this not just be an artifact of imaging a 3D IB, so of course there’s more RNA at the center because the IB is thicker at the center? This could be examined in the confocal z stacks that were acquired according to materials and methods

We thank the reviewer for this comment and now present a more convincing image in Fig. 1. We acknowledge that the distribution of the RNA or proteins within IBs and IBAGs is more evident from 3D images and therefore now provide 3D data for the 5EU staining corresponding to Fig.1 (Supplementary Fig.1) and for M2-1mGFP in IBAGs (Supplementary Fig.6).

Images of smooth staining in IBs under confocal microscopy are visible in Figure 6.

Figure 2: Here, (unlike Figure 1), the concentrated spots (named IBAGs by the authors) are

observable, but again here, at confocal resolution, statements like “genomic RNA signal was present in the whole IBs but being more abundant at the periphery” are hard to defend. (Also “being” here is grammatically incorrect).

Figure 2 has been modified. The staining pattern of the viral genome clearly contrasts with that of the mRNA.

We replaced “...signal was present in the whole IBs but being more abundant...” by “...signal was present in the whole IBs but **was** more abundant...”

Page 6: “The main obstacle for studying the localization of proteins inside IBs is that immunostaining of N or P leads to a fluorescent signal restricted to the periphery of the IBs...” how have the authors ascertained that this sort of artifact is not also a problem for RNA labeling?

We fully agree with the reviewer’s remark. This ring staining of IBs remains mysterious, but only concerns N and P proteins. Please see our response to reviewer 2 who pointed out the same issue (point 2).

Figure 5: The difference between N, L, and P localization vs M21 localization is impressive. A sentence in the text pointing this out would add weight to the claims that there is exclusion of N, L, P from IBAGs.

We modified the results section to highlight the clear differences between M2-1 and the other proteins locations.

The revised text now reads:

“The super resolution images reveal in details the architecture of the IBs. M2-1 clearly colocalize with poly(A) spots in IBAGs while the N, P and L proteins are present all throughout the IBs volume except in the IBAGs. Moreover they allowed us...”

Instead of:

“The super-resolution images clearly confirmed the exclusion of L, P, and N from IBAGs and allowed us to...”

Figure 5: The authors should mention in the transition from Figure 4 to Figure 5 that the PALM/STORM experiments in STORM buffer are no longer live-cell experiments (vs. Figure 4).

The images in Figure 4 were also obtained on fixed cells since FISH staining requires cell fixation. The location of fluorescent proteins was not modified by fixation (see Supplementary Fig.10)

Other modifications of the manuscript

Subheadings of the results and methods sections were modified to meet Nature Communications format requirements (less than 60 characters)

“Viral mRNA but not genomic RNA concentrates in IB associated granules” was replaced by:
“Viral mRNA specifically concentrate in IB associated granules”

“M2-1 concentrates in IBAGs from which N, P and L are excluded” was replaced by:
“M2-1 concentrates in IBAGs from which N, P and L are absent”

“IBAGs are highly dynamic structures transiently storing viral mRNA” was replaced by:
“IBAGs are dynamic structures transiently storing viral mRNA”

“Cell infection, transfection and minigenome replication assay” was replaced by:
“Cell infection, transfection and minigenome assay”

Data availability section was added at the end of the methods section

Data Availability

The nucleotide sequence of RSV-M2-1mGFP is available in the Genbank nucleotide database with accession code KX348546

The scale bars labelling was removed from the figures and added in the legends.

We also improved the wording in several places as can be seen in the revised manuscript using the change tracking feature.

REVIEWERS' COMMENTS:

Reviewer #2 (Remarks to the Author):

The authors have addressed the concerns adequately.

A minor remaining issue relates to Fig. 2 and a statement on the distribution of genomic RNA. From the images (including the new images) it appears that genomic RNA is abundant throughout the IB (not only abundant in periphery as the authors state), except in IBAGs. In the chosen images there is a large IBAG in the center and hence genomic RNA is not at the center; However, the IBAGs can locate to various places within the IB, in part presumably because they need to release their contents to the cytosol. Thus, genomic RNA is absent from IBAGs, and mRNAs are present in IBAGs at high levels. How genomic RNA distributes within the IB (outside of IBAGs) is not clear and no evidence is brought to show preferred distribution. Since the latter is not a question pursued in this study, this does not negatively impact its major findings.

Reviewer #3 (Remarks to the Author):

My concerns have been addressed satisfactorily. This is an exciting study that advances the field.

Reviewer #4 (Remarks to the Author):

I appreciate the authors' responses to my comments and that they have replaced certain figure panels with new selections that more directly demonstrate their points. Their statements interpreting these figures are more clear now. I'm still a bit uncomfortable with reliance on images of partially functional fluorescent protein fusions, but since the super-resolution images are but one piece of the evidence presented here, and since there has never before been direct evidence of viral RNA synthesis in IBs, the impact of the paper is obvious. I have no further concerns.

RESPONSE TO REVIEWERS' COMMENTS:

Reviewer #2 (Remarks to the Author):

The authors have addressed the concerns adequately.

A minor remaining issue relates to Fig. 2 and a statement on the distribution of genomic RNA. From the images (including the new images) it appears that genomic RNA is abundant throughout the IB (not only abundant in periphery as the authors state), except in IBAGs. In the chosen images there is a large IBAG in the center and hence genomic RNA is not at the center; However, the IBAGs can locate to various places within the IB, in part presumably because they need to release their contents to the cytosol. Thus, genomic RNA is absent from IBAGs, and mRNAs are present in IBAGs at high levels. How genomic RNA distributes within the IB (outside of IBAGs) is not clear and no evidence is brought to show preferred distribution. Since the latter is not a question pursued in this study, this does not negatively impact its major findings.

We think there is a misunderstanding due to our inappropriate usage of the word “periphery”. We wanted to say that genomic RNA was distributed in the whole IB except for the IBAGs just as underlined by reviewer 2.

We modify the text accordingly,

“However, unlike viral mRNAs, the genomic RNA signal was present **throughout the whole IBs except the IBAGs.**” instead of “However, unlike viral mRNAs, the genomic RNA signal was present in the whole IBs, but was more abundant at the periphery of the IBs.”

Reviewer #3 (Remarks to the Author):

My concerns have been addressed satisfactorily. This is an exciting study that advances the field.

Reviewer #4 (Remarks to the Author):

I appreciate the authors' responses to my comments and that they have replaced certain figure panels with new selections that more directly demonstrate their points. Their statements interpreting these figures are more clear now. I'm still a bit uncomfortable with reliance on images of partially functional fluorescent protein fusions, but since the super-resolution images are but one piece of the evidence presented here, and since there has never before been direct evidence of viral RNA synthesis in IBs, the impact of the paper is obvious. I have no further concerns.

We thank again all the reviewers for their constructive comments and their suggestions that have improved our work.